# PI3K/HSCB axis facilitates FOG1 nuclear translocation to promote erythropoiesis and megakaryopoiesis

Gang Liu[1]*[†], Yunxuan Hou[1][†], Xin Jin[1], Yixue Zhang[1], Chaoyue Sun[1], Chengquan Huang[1], Yujie Ren[1], Jianmin Gao[2], Xiuli Wang[3], Xiumei Jiang[2]*

[1]Key Laboratory of Molecular Epigenetics of the Ministry of Education, Northeast Normal University, Changchun, China; [2]School of Chemistry, Northeast Normal University, Changchun, China; [3]School of Life Science, Northeast Normal University, Changchun, China

*For correspondence:
liug268@126.com (GL);
jiangxm161@nenu.edu.cn (XJ)

[†]These authors contributed equally to this work

**Competing interest:** The authors declare that no competing interests exist.

**Abstract** Erythropoiesis and megakaryopoiesis are stringently regulated by signaling pathways. However, the precise molecular mechanisms through which signaling pathways regulate key transcription factors controlling erythropoiesis and megakaryopoiesis remain partially understood. Herein, we identified heat shock cognate B (HSCB), which is well known for its iron–sulfur cluster delivery function, as an indispensable protein for friend of GATA 1 (FOG1) nuclear translocation during erythropoiesis of K562 human erythroleukemia cells and cord-blood-derived human CD34+CD90+hematopoietic stem cells (HSCs), as well as during megakaryopoiesis of the CD34+CD90+HSCs. Mechanistically, HSCB could be phosphorylated by phosphoinositol-3-kinase (PI3K) to bind with and mediate the proteasomal degradation of transforming acidic coiled-coil containing protein 3 (TACC3), which otherwise detained FOG1 in the cytoplasm, thereby facilitating FOG1 nuclear translocation. Given that PI3K is activated during both erythropoiesis and megakaryopoiesis, and that FOG1 is a key transcription factor for these processes, our findings elucidate an important, previously unrecognized iron–sulfur cluster delivery independent function of HSCB in erythropoiesis and megakaryopoiesis.

## eLife assessment

This **fundamental** work significantly advances our understanding of how FOG1 nuclear localization is regulated during erythropoiesis and megakaryopoiesis, including the role of EPO and MPL/TPO signaling in this process. The authors provide **compelling** evidence using both K562 and CD34+ cells that heat shock cognate B (HSCB) can promote the proteasomal degradation of TACC3 to regulate the nuclear localization of FOG1, and that this function is independent of its role in iron-sulfur cluster (ISC) biogenesis. Together these data will be of interest to the fields of hematopoiesis and cell biology.

## Introduction

Erythropoiesis and megakaryopoiesis, respectively, refer to the processes through which hematopoietic stem cells (HSCs) differentiate into mature red blood cells (RBCs) and megakaryocytes/platelets. Given the physiological significance of RBCs, megakaryocytes, and platelets, it is not surprising that erythropoiesis and megakaryopoiesis are tightly regulated by a myriad of molecular mechanisms. Erythropoietin (EPO) is a fundamental cytokine promoting erythropoiesis via binding with erythropoietin receptor (EPOR) (*Kuhrt and Wojchowski, 2015*). Converging evidence

indicates that the EPO/EPOR signaling is crucial for the proliferation and survival of colony-forming unit erythroid cells (CFU-Es) in vitro (*Nandakumar et al., 2016*). Although the EPO/EPOR signaling was dispensable for in vivo production of early committed erythroid progenitors, *Epo*$^{-/-}$ and *EpoR*$^{-/-}$ mice exhibited embryonic death due to deficient definitive erythropoiesis (*Wu et al., 1995*). These studies highlight the indispensability of EPO/EPOR signaling during erythropoiesis. Thrombopoietin (TPO) and its receptor myeloproliferative leukemia protein (MPL) constitute the primary regulator of megakaryopoiesis (*Hitchcock et al., 2021*). Studies indicated that TPO alone could induce the production of megakaryocyte colony-forming units (CFU-MKs), while addition of other cytokines including stem cell factor (SCF), interleukin 3 (IL3), and IL11 markedly increased the size, but not the number, of these colonies (*Kaushansky et al., 1995*; *Kaushansky et al., 1994*). The biological significance of TPO/MPL signaling pathway was also evidenced by the development of thrombocytopenia in Mpl-deficient mice as well as in human subjects carrying *MPL* and *THPO* mutations (*Gurney et al., 1994*; *Noris and Pecci, 2017*). Both EPOR and MPL belong to the type-I family of cytokine receptors. They do not possess kinase activity per se, but can transphosphorylate Janus tyrosine kinase 2 (JAK2), which in turn activates several downstream signaling cascades, including the JAK2/signal transducers and activators of transcription (STAT), phosphoinositol-3-kinase (PI3K)/ AKT and rat sarcoma (RAS)/RAF/mitogen-activated protein kinase (MAPK) pathways (*Eggold and Rankin, 2019*; *Hitchcock et al., 2021*), all of which contribute more or less to both erythropoiesis and megakaryopoiesis.

The essence of differentiation is the generation of cells with specific functions, which is realized by the differentiation stage-dependent transcriptional activation and suppression of innumerable genes. This is why there is a generally accepted notion that transcription factors (TFs) are the main drivers of differentiation (*Lambert et al., 2018*). Each of the differentiation processes of erythropoiesis and megakaryopoiesis is controlled by a combination of key TFs, including GATA1, friend of GATA1 (FOG1), KLF1, TAL1, MEIS1, etc., for the erythroid lineage (*Doré and Crispino, 2011*; *Kwon et al., 2021*), and GATA1, FOG1, TAL1, MEIS1, FLI1, etc., for the megakaryocytic lineage (*Doré and Crispino, 2011*; *Kwon et al., 2021*). Since cell signaling is the dominant way by which erythroid and megakaryocytic progenitors (or their common progenitors) sense differentiation signals, it is crucial to understand how these key erythroid and megakaryocytic TFs are regulated by signaling pathways, especially those downstream of EPO/EPOR and TPO/MPL signaling. However, to date, research on this topic remains scarce. A study deepened our understanding about this issue by demonstrating that the PI3K–AKT pathway can phosphorylate GATA1 at serine 310 in erythroid cells, thereby augmenting its transcription activity (*Zhao et al., 2006*). However, given that the proper functioning of GATA1 in regulating target gene expression relies on its binding with a specific cofactor (*Gutiérrez et al., 2020*), how EPO/EPOR and TPO/MPL signaling pathways regulate the activity of GATA1-binding partners, such as FOG1, warrants further investigation.

Heat shock cognate B (HSCB or HSC20) is widely known for its function during the delivery of nascent iron–sulfur clusters (ISCs) to recipient proteins (*Maio et al., 2014*). The biogenesis of ISCs starts with the assembly of nascent ISCs on a multi-protein complex composed of iron–sulfur cluster assembly enzyme (ISCU), nitrogen fixation 1 homolog (NFS1), LYR motif containing 4 (LYRM4 or ISD11), and acyl carrier protein (ACP) (*Maio et al., 2020*). After the ISCs are synthesized, ISCU binds with an ISC delivery chaperone/cochaperone complex comprising HSCB and HSPA9 to transfer the nascent ISCs to target Fe–S proteins or secondary ISC carriers (*Maio et al., 2020*). Recently, *Crispin et al., 2020* described a female patient who suffered from congenital sideroblastic anemia (CSA) due to heterozygous variations that resulted in decreased expression of HSCB. In addition to the anemic phenotype, the proband also developed moderate thrombocytopenia and mild neutropenia. Consistent with the proband's clinical presentations, *Hscb*$^{fl/-}$ mice with a *Vav1-Cre* transgene also exhibited reduced RBC and platelet counts (*Crispin et al., 2020*). Based on these findings, we speculated that HSCB may be implicated with hematopoiesis, especially erythropoiesis and megakaryopoiesis. To test this hypothesis, herein we explored the function of HSCB during erythropoiesis and megakaryopoiesis. Our findings indicated that HSCB can be phosphorylated by PI3K to promote FOG1 nuclear translocation during human erythropoiesis and megakaryopoiesis. This study describes a previously unrecognized iron–sulfur cluster delivery independent function of HSCB, thereby providing new insights into human erythropoiesis and megakaryopoiesis.

## Results

### HSCB exerted an ISC delivery independent function during erythropoiesis of HSCs

First, we investigated if HSCB is implicated in erythropoiesis by inhibiting its expression in CD34+CD90+HSCs via a mixture of shRNAs. We also knocked down the expression of human *ISCU* and *NFS1* genes in these cells to determine the impact of ISC biogenesis deficiency on erythropoiesis. The high-purity CD34+CD90+HSC population was first obtained (*Figure 1—figure supplement 1* and *Figure 1—figure supplement 2A*) and divided into five groups of Control, shNC, shHSCB, shISCU, and shNFS1. HSCs in the Control group were untransfected, whereas those in the other four groups were transfected with the corresponding shRNA plasmids. The HSCs in these groups were induced for erythropoiesis for 17 days to yield erythrocytes at day 17 of differentiation (Ery 17 days) and collected for analyses (*Figure 1—figure supplement 1*). *Figure 1A* demonstrates the efficiency of shRNA knockdown maintained throughout the experimental period. The activities of aconitase and pyruvate dehydrogenase (PDH), which depend on intact ISC biogenesis (*Cameron et al., 2011*; *Castro et al., 2019*), decreased significantly in the shHSCB, shISCU, and shNFS1 groups compared with the shNC group (all p values were less than 0.05 or 0.01 in Student's $t$ test, *Figure 1B, C*), demonstrating that ISC biogenesis was strongly inhibited in these groups. Since heme biosynthesis is regulated by ISC biogenesis at several nodes (*Liu et al., 2020*), we further explored whether heme biosynthesis was also inhibited in the shHSCB, shISCU, and shNFS1 groups. Indeed, these three groups exhibited significantly lower heme contents (all p values were less than 0.05 or 0.01 in Student's $t$ test, *Figure 1D*) and paler cell pellets (*Figure 1—figure supplement 2B*) than the shNC group. Surprisingly, among the groups subjected to shRNA plasmids transfection, only the shHSCB group displayed significantly inhibited erythropoiesis, as evidenced by the reduced proportion of CD71+CD235a+progenitors (p < 0.001, Student's $t$ test, *Figure 1E, F*) and the decreased mRNA expression levels of erythroid marker genes *HBB*, *ALAS2*, and *GYPC* (all p values were less than 0.01 or 0.001 in Student's $t$ test, *Figure 1G*). We then purified the CD71−CD235a− subpopulation from the shHSCB group of Ery 17 days via cell sorting and characterized the differential status of these cells. As shown in *Figure 1H*, the majority of CD71−CD235a− cells from the shHSCB group were CD34+CD38+CD45RA−IL3RA−, exhibiting a surface marker expression pattern (CD34+CD38+CD45RA−IL3RA−CD71−CD235a−) resembling that observed on CD34+CD90+HSCs induced for erythropoiesis for 2 days (E 2-day HSCs) (*Figure 1—figure supplement 2C* and Figure 5C). These findings revealed that HSCB knockdown detained the erythropoiesis of HSCs at an early stage and that HSCB might be involved in key events regulating early erythropoiesis at this specific stage. Therefore, E 2-day HSCs were characterized in following experiments to explore the molecular mechanisms through which HSCB deficiency impeded HSCs erythropoiesis. Altogether, these data indicated that HSCB knockdown impaired both heme biosynthesis and erythropoiesis in erythroid progenitors, while knockdown of ISC assembly proteins markedly reduced heme production but did not significantly affect erythropoiesis, suggesting an ISC delivery independent function of HSCB during early erythropoiesis of CD34+CD90+HSCs.

### HSCB deficiency inhibited the erythropoiesis of K562 cells

K562 erythroleukemia cells are widely employed to study the molecular events implicated with human erythropoiesis (*Nandakumar et al., 2016*). To verify the function of HSCB in erythropoiesis and explore the underlying molecular mechanisms, we established *HSCB*-knockout K562 cells (CRISPR HSCB) and induced erythroid differentiation of these cells with EPO (EPO+CRISPR HSCB). After fetal bovine serum (FBS) deprivation treatment (see the Materials and methods section for details), untransfected K562 cells without EPO treatment, those only treated with EPO and those first transfected with the negative control CRISPR plasmid and then treated with EPO, respectively, constituted the Untreated, EPO, and EPO+CRISPR NC groups. As revealed by *Figure 2A–C*, compared with the Untreated group, the EPO, and EPO+CRISPR NC groups exhibited significantly higher levels of erythrocyte membrane proteins GYPA and SPTA1 (all p values were less than 0.05 or 0.01 in Student's $t$ test), significantly increased heme contents (all p values were less than 0.01 or 0.001 in Student's $t$ test) and elevated mRNA expression levels of erythroid-specific genes *HBB*, *ALAS2*, and *GYPC*, indicating that the EPO treatment successfully induced the erythropoiesis of K562 cells. In the EPO+CRISPR HSCB group, however, the protein levels of GYPA and SPTA1, heme content and mRNA levels of *HBB*, *ALAS2*, and *GYPC* all decreased markedly compared with those of the EPO+CRISPR NC group (all p values were

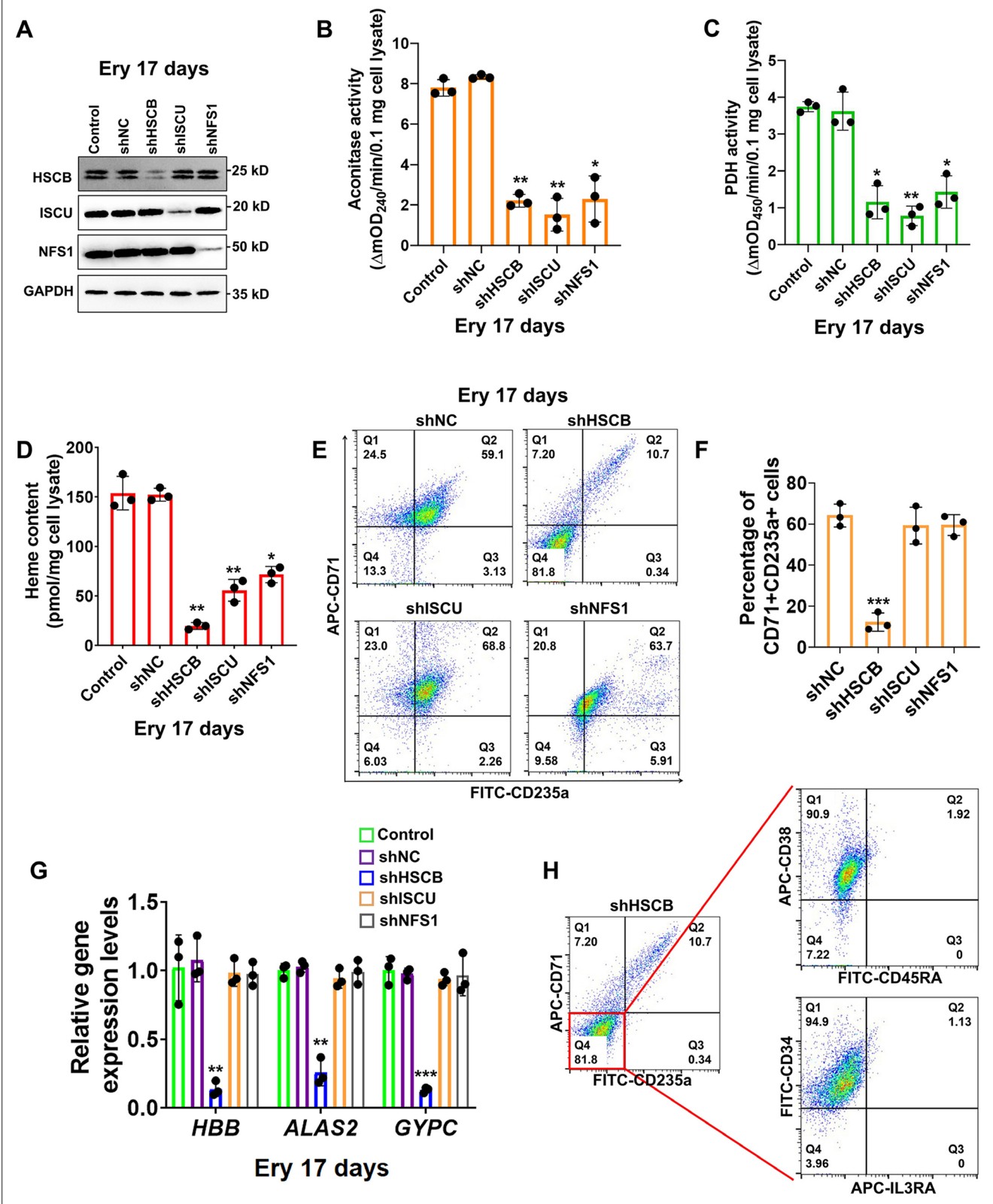

**Figure 1.** Heat shock cognate B (HSCB) played an important iron–sulfur cluster (ISC) delivery independent function during erythropoiesis of CD34+CD90+hematopoietic stem cells (HSCs). (**A**) Western blotting analyses on Ery 17 days verified the efficacy of short hairpin RNA (shRNA) knockdown maintained throughout the experimental period. (**B**) Aconitase activity, (**C**) pyruvate dehydrogenase (PDH) activity, and (**D**) heme content of the shHSCB, shISCU, and shNFS1 groups were significantly lower than those of the shNC group, indicating inhibited ISC biogenesis in the shHSCB,

*Figure 1 continued on next page*

*Figure 1 continued*

shISCU, and shNFS1 groups. (**E**) Representative flow cytometry scatter plots and (**F**) statistics of three biologically independent flow cytometry assays displaying the percentages of CD71+CD235a+progenitors in the shNC, shHSCB, shISCU, and shNFS1 groups. (**G**) Quantitative real-time PCR (qRT-PCR) analyses of the mRNA levels of *HBB*, *ALAS2*, and *GYPC* genes for the shNC, shHSCB, shISCU, and shNFS1 groups. The flow cytometry and qRT-PCR results, respectively, signified significantly decreased percentages of CD71+CD235a+progenitors and significantly lower mRNA levels of the erythroid-specific genes in the shHSCB group compared with the shNC group, revealing inhibited erythropoiesis of the shHSCB group of Ery 17 days. (**H**) Flow cytometry scatter plots showing that the majority of CD71−CD235a− cells from the shHSCB group were CD34+CD38+CD45RA−IL3RA−. Error bars denote standard deviations; *, **, and ***, respectively, denote p values less than 0.05, 0.01, and 0.001 relative to the shNC groups, two-sided Student's *t* test.

The online version of this article includes the following source data and figure supplement(s) for figure 1:

**Source data 1.** Labeled and raw Western blots for *Figure 1A*.

**Figure supplement 1.** A schematic diagram for work flows of gene knockdown and overexpression experiments in CD34+CD90+HSCs and the downstream progenitors.

**Figure supplement 2.** Characterization of CD34+CD90+HSCs before and after induction for erythropoiesis.

less than 0.01 or 0.001 in Student's *t* test), demonstrating that HSCB knockout hindered the erythropoiesis of K562 cells induced by EPO. Erythropoiesis is underpinned by transcriptional changes mainly driven by nuclear proteins (*Doré and Crispino, 2011*; *Nandakumar et al., 2016*). To explore why HSCB silencing inhibited the erythropoiesis of K562 cells induced by EPO, we compared the nuclear proteomic profiles of the EPO+CRISPR NC and EPO+CRISPR HSCB groups via performing tandem mass tag-based mass spectrometry (TMT-MS) on the nuclear fractions. As shown in *Figure 2D*, nine downregulated and two upregulated differentially expressed nuclear proteins (DENPs, see the Materials and methods section for the definition) were identified in the EPO+CRISPR HSCB group. Among these 11 DENPs, only NAA30, LYAR, and FOG1 were *bona fide* nuclear proteins (*Figure 2E*). We further tested whether these three DENPs were indeed differentially expressed in the nuclear fractions of K562 cells and E 2-day HSCs after HSCB silencing. Western blotting (WB) analyses revealed that FOG1 was the only nuclear protein expressed at significantly different levels in both K562 cells and E 2-day HSCs upon HSCB deficiency (p values were less than 0.05 or 0.01 in Student's *t* test, *Figure 2F, G*). We also tested the nuclear abundance of GATA1 in these cells but observed no significant changes (*Figure 2F, G*). These data confirmed that the nuclear abundance of FOG1 was decreased in HSCB-deficient EPO-induced K562 cells and E 2-day HSCs.

## HSCB deficiency impaired the erythropoiesis of K562 cells and HSCs by prohibiting FOG1 nuclear translocation

Given the importance of FOG1 during human erythropoiesis, we determined whether the decreased nuclear abundance of FOG1 was responsible for the inhibited erythropoiesis in HSCB-deficient EPO-induced K562 cells and CD34+CD90+HSCs. We overexpressed human FOG1 protein in the EPO+CRISPR HSCB K562 cells and the shHSCB CD34+CD90+HSCs to, respectively, establish the EPO+CRISPR HSCB+FOG1 OE and shHSCB+FOG1 OE groups. As shown in *Figure 3A–E*, FOG1 overexpression counteracted the inhibitory effects of HSCB silencing on the erythropoiesis of K562 cells and CD34+CD90+HSCs because, except for the heme contents, the EPO+CRISPR HSCB+FOG1 OE and shHSCB+FOG1 OE groups exhibited comparable levels of all the parameters evaluated relative to their corresponding control groups. Interestingly, the total protein level of GATA1 was not significantly affected by HSCB deficiency or FOG1 overexpression in EPO-induced K562 cells and CD34+CD90+HSCs (*Figure 3A*). Furthermore, the abundance of FOG1 in total cell lysates was not significantly affected by HSCB deficiency neither (*Figure 3A*). These data verified that HSCB deficiency hindered the erythropoiesis of K562 cells and CD34+CD90+HSCs by reducing the nuclear abundance of FOG1. To gain more insights into the molecular mechanisms by which HSCB deficiency decreased FOG1 nuclear abundance, we also measured cytoplasmic FOG1 protein levels in EPO+CRISPR HSCB K562 cells and the shHSCB group of E 2-day HSCs. We observed that these cells expressed significantly higher cytoplasmic levels of FOG1 than their corresponding controls (all p values were less than 0.01 or 0.001 in Student's *t* test, *Figure 3F*), indicating that HSCB deficiency prohibited FOG1 nuclear translocation. These findings were further confirmed by immunofluorescence assays on K562 cells, in which the red fluorescence of FOG1 in the EPO+CRISPR NC group was mostly observed in the nuclei and that in the EPO+CRISPR HSCB group was mainly found in the cytoplasm (*Figure 3G*). Collectively,

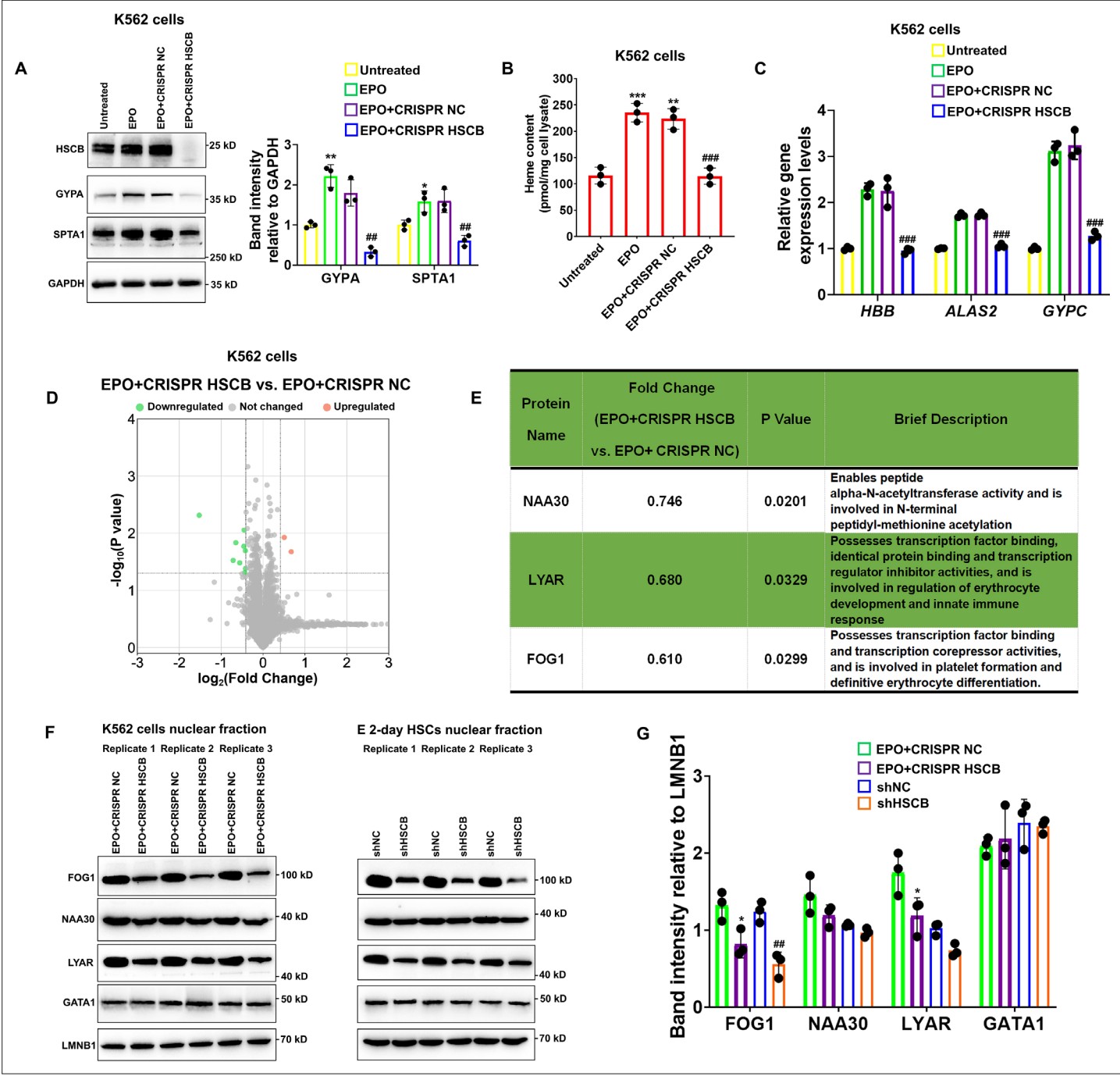

**Figure 2.** Knockout of heat shock cognate B (HSCB) with the CRISPR–Cas9 technique impaired erythropoietin (EPO)-induced erythropoiesis of K562 cells. (**A**) Western blotting analyses exhibited successful knockout of HSCB and decreased levels of erythrocyte membrane proteins (GYPA and SPTA1) in the EPO+CRISPR HSCB K562 cells. Error bars denote standard deviations; * and **, respectively, signify p values less than 0.05 and 0.01 compared with the Untreated group, while ## means p < 0.01 relative to the EPO+CRISPR NC group, two-sided Student's *t* test. (**B**) Heme contents and (**C**) mRNA levels of the erythroid-specific genes were significantly lower in the EPO+CRISPR HSCB group than in the EPO+CRISPR NC group. Error bars denote standard deviations; ** and ***, respectively, represent p values less than 0.01 and 0.001 compared with the Untreated group, while ### means p < 0.001 relative to the EPO+CRISPR NC group, two-sided Student's *t* test. (**D**) A volcano plot for the differentially expressed nuclear proteins (DENPs) detected by tandem mass tag-based mass spectrometry (TMT-MS) assays and (**E**) detailed information of the *bona fide* DENPs between the EPO+CRISPR NC and EPO+CRISPR HSCB groups. (**F**) Western blotting and (**G**) the corresponding band intensity analyses for FOG1, NAA30, LYAR, and GATA1 proteins in the nuclear fractions of the EPO+CRISPR NC and EPO+CRISPR HSCB K562 cells and shNC and shHSCB E 2-day HSCs. Friend of GATA1 (FOG1) was identified as the only nuclear protein expressed at significantly different levels in both K562 cells and E 2-day HSCs after HSCB knockout or knockdown.

*Figure 2 continued on next page*

*Figure 2 continued*

Error bars denote standard deviations; * represents p < 0.05 relative to the EPO+CRISPR NC group, while [##] means p < 0.01 compared with the shNC group, two-sided Student's *t* test.

The online version of this article includes the following source data for figure 2:

**Source data 1.** Labeled and raw Western blots for *Figure 2A, F*.

these observations indicated that HSCB deficiency impaired the erythropoiesis of K562 cells and HSCs by prohibiting FOG1 nuclear translocation.

## HSCB deficiency prohibited FOG1 nuclear translocation by enhancing its interaction with TACC3

Nuclear translocation of a specific protein is ultimately regulated by protein-protein interactions (*Cautain et al., 2015*; *Wing et al., 2022*). Therefore, to further explain how HSCB deficiency prohibited FOG1 nuclear translocation, we next sought to identify differential binding proteins (DBPs, see the Materials and methods section for the definition) through affinity purification-MS analyses. We identified two DBPs, namely MYO1E and TACC3 (*Figure 4A*). Their binding affinities with FOG1 increased significantly after HSCB expression was silenced (p values were less than 0.05 in Student's *t* test). We focused only on TACC3 because MYO1E did not bind with FOG1 in either the shNC or shHSCB group of E 2-day HSCs (*Figure 4—figure supplement 1*). We first performed co-immunoprecipitation (co-IP) experiments to test whether FOG1 indeed exhibited increased binding affinities with TACC3 in EPO-induced K562 cells and E 2-day HSCs after HSCB silencing. As shown in *Figure 4B, C*, the interaction between FOG1 and TACC3 was only detectable in HSCB-deficient cells, confirming that HSCB deficiency promoted the interaction between FOG1 and TACC3. KHS101, a TACC3 inhibitor, can both reduce the activity of TACC3 and decrease its protein abundance (*Polson et al., 2018*). Therefore, we tested whether KHS101 treatment could decrease the protein level of TACC3 in these cells and thereby alleviate the inhibitory effect of HSCB deficiency on their erythropoiesis. Indeed, as shown in *Figure 4D–H*, KHS101 treatment enhanced FOG1 nuclear translocation [as reflected by the significantly increased relative nuclear FOG1 (N-FOG1) to cytoplasmic FOG1 (C-FOG1) ratios in the KHS101 treatment groups (p values were less than 0.05 or 0.01 in Student's *t* test)] and restored erythropoiesis in EPO+CRISPR HSCB K562 cells and shHSCB HSCs. Altogether, these data exhibited that HSCB deficiency prohibited FOG1 nuclear translocation by promoting its interaction with TACC3, and that reducing TACC3 abundance by KHS101 treatment alleviated the inhibitory effect of HSCB deficiency on the erythropoiesis of K562 cells and CD34+CD90+HSCs.

## HSCB facilitated FOG1 nuclear translocation by binding with and mediating the proteasomal degradation of TACC3 upon activation of the EPO/EPOR signaling

To further understand how HSCB facilitated FOG1 nuclear translocation and why HSCB deficiency enhanced the protein level of TACC3, we transfected an HSCB-overexpressing plasmid into K562 cells and treated the transfected K562 cells with EPO to establish an EPO+HSCB OE group. As shown in *Figure 5A*, EPO treatment significantly reduced the protein level of TACC3 and promoted FOG1 nuclear translocation (all p values were less than 0.05 or 0.01 in Student's *t* test). HSCB overexpression further decreased the protein level of TACC3 and enhanced FOG1 nuclear abundance. However, after the EPO+HSCB OE K562 cells were treated with MG132, a proteasome inhibitor, the protein level of TACC3 and the nuclear abundance of FOG1 returned to those in untreated K562 cells. Co-IP assays revealed that TACC3 interacted with HSCB when the K562 cells were treated with EPO, whereas it interacted with FOG1 in the untreated and EPO+HSCB OE + MG132 K562 cells (*Figure 5B*). These data indicated that, upon EPO treatment, HSCB could bind with and mediate the proteasomal degradation of TACC3, thereby preventing TACC3 from binding with FOG1 in K562 cells. We further characterized the protein expression profiles of HSCB, cytoplasmic/nuclear FOG1 and TACC3, as well as their physical interactions, at different stages during erythropoiesis of CD34+CD90+HSCs. The cells were induced for erythropoiesis for 0, 2, 4, 10, 17, and 24 days and harvested for analyses (*Figure 5C*). As shown in *Figure 5D, E*, the abundance of TACC3 decreased in untreated CD34+CD90+HSCs but exhibited not obvious change in their HSCB-deficient counterparts during the first 4

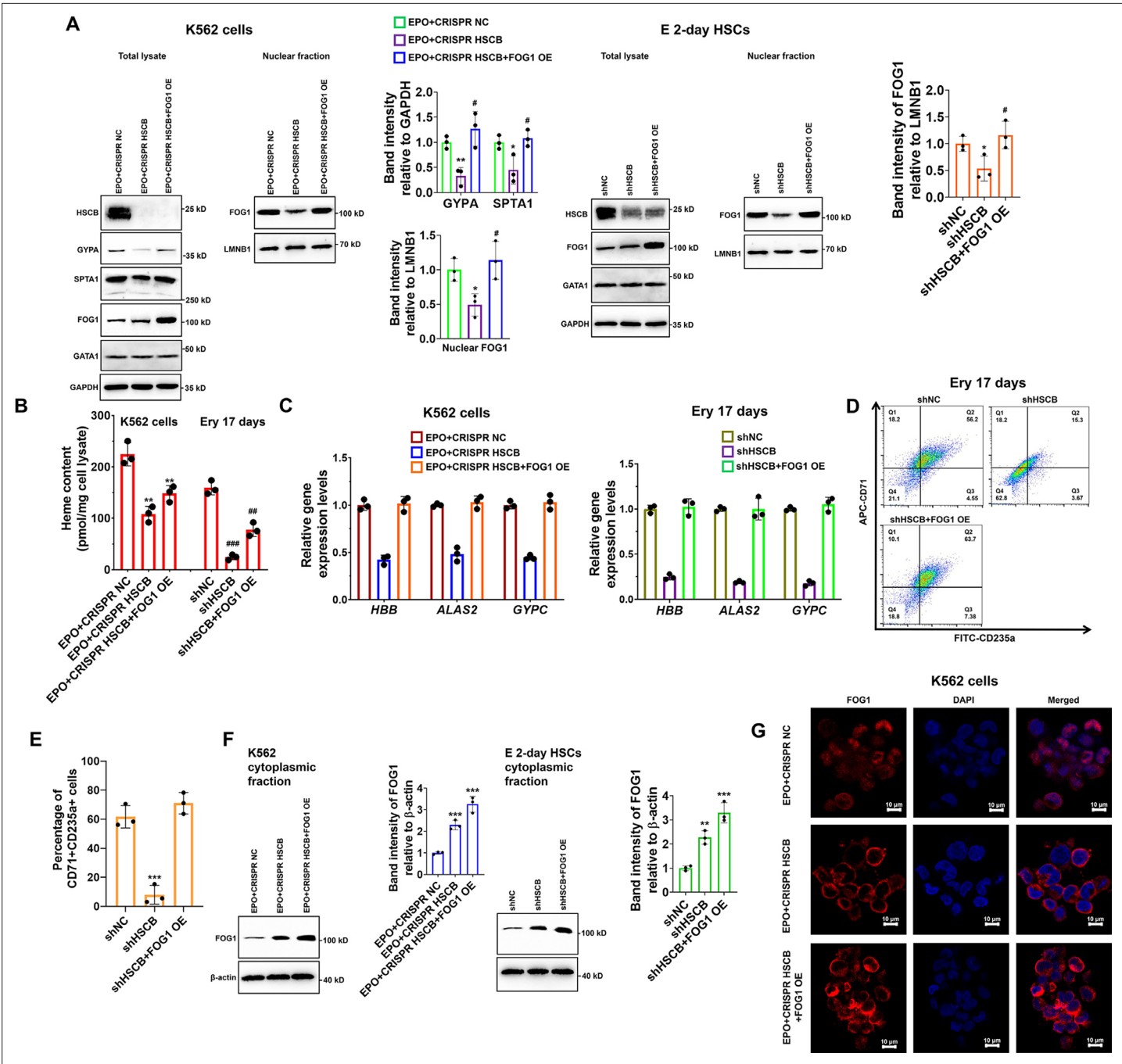

**Figure 3.** Impaired FOG1 nuclear translocation was responsible for the inhibited erythropoiesis of heat shock cognate B (HSCB)-deficient K562 cells and hematopoietic stem cells (HSCs). (**A**) Western blotting analyses indicated that FOG1 overexpression increased the levels of nuclear FOG1 in HSCB-deficient erythropoietin (EPO)-induced K562 cells and E 2-day HSCs, as well as the levels of erythrocyte membrane proteins in HSCB-deficient EPO-induced K562 cells without significantly affecting the expression level of GATA1. Error bars denote standard deviations; * and **, respectively, represent p values less than 0.05 and 0.01 compared with the EPO+CRISPR NC or shNC group, while # denotes p < 0.05 relative to the EPO+CRISPR HSCB or shHSCB group, two-sided Student's *t* test. (**B**) Heme contents in the EPO+CRISPR NC, EPO+CRISPR HSCB and EPO+CRISPR HSCB+FOG1 OE groups of K562 cells, as well as in the shNC, shHSCB, and shHSCB+FOG1 OE groups of Ery 17 days. Error bars denote standard deviations; ** denotes p < 0.01 relative to the EPO+CRISPR NC group, while ## and ###, respectively, represent p values less than 0.01 and 0.001 compared with the shNC group, two-sided Student's *t* test. (**C**) qRT-PCR analyses on HSCB-deficient EPO-induced K562 cells and HSCB-deficient Ery 17 days demonstrated resumed mRNA expression levels of erythroid-specific genes after FOG1 overexpression. (**D**) Representative flow cytometry scatter plots and (**E**) statistics of three biologically independent flow cytometry assays showing that the percentages of CD71+CD235a+progenitors among HSCB-deficient Ery 17 days recovered after FOG1 overexpression. Error bars denote standard deviations; *** signifies p < 0.001 relative to the shNC group, two-sided

*Figure 3 continued on next page*

*Figure 3 continued*

Student's *t* test. (**F**) Western blotting analyses of cytoplasmic FOG1 and (**G**) immunofluorescence (IF) assays for FOG1 revealed inhibited FOG1 nuclear translocation in HSCB-deficient EPO-induced K562 cells and E 2-day HSCs (scale bars = 10 µm). Error bars denote standard deviations; ** and *** in panel (**F**), respectively, represent p values less than 0.01 and 0.001 compared with the EPO+CRISPR NC or shNC group, two-sided Student's *t* test.

The online version of this article includes the following source data for figure 3:

**Source data 1.** Labeled and raw Western blots for *Figure 3A, F*.

days of erythropoiesis, indicating that the decreased TACC3 protein abundance during early erythroblast differentiation was mainly due to the effect of HSCB. Additionally, the nuclear abundance of FOG1 and the interaction between HSCB and TACC3 all gradually increased from days 0 to 4, resembling the changing trend of the number of EPOR+ cells (*Figure 5D, F, G*). These findings were consistent with those observed in K562 cells and further highlighted that the EPO/EPOR signaling was indispensable for HSCB to bind with and mediate the proteasomal degradation of TACC3 to facilitate FOG1 nuclear translocation (hereinafter defined as HSCB functionalization).

## Phosphorylation of HSCB by PI3K was necessary for its functionalization during human erythropoiesis

To determine which downstream cascade of EPO/EPOR signaling was important for HSCB functionalization, we treated K562 cells with EPO plus ruxolitinib, wortmannin, and lonafarnib to, respectively, inhibit the activities of JAK2, PI3K, and RAS upon activation of the EPO/EPOR signaling. As shown in *Figure 6—figure supplement 1*, treatments with ruxolitinib, wortmannin, and lonafarnib, respectively, decreased the phosphorylation levels of STAT5, AKT, and MEK1/2 in EPO-induced K562 cells, confirming that these treatments were effective. Among the inhibitors, ruxolitinib and wortmannin significantly ameliorated the protein level of TACC3 and diminished the nuclear translocation of FOG1 (all p values were less than 0.05 or 0.001 in Student's *t* test, *Figure 6A*), suggesting that JAK2 and PI3K activities are both necessary for HSCB functionalization in EPO-induced K562 cells. Since PI3K functions downstream of JAK2 in the EPO/EPOR signaling, we then focused only on PI3K to investigate its role in HSCB functionalization. Consistent with the WB findings, our co-IP assays demonstrated that inhibition of PI3K activity by wortmannin abolished the interaction between HSCB and TACC3 and thereby allowed TACC3 to bind with FOG1 in EPO-induced K562 cells and E 2-day HSCs (*Figure 6B*), implying that the HSCB–TACC3 and FOG1–TACC3 interactions may be mutually exclusive. Further analysis of protein phosphorylation status via co-IP assays revealed that EPO treatment enhanced the overall phosphorylation level of cytosolic HSCB, but not that of TACC3, in K562 cells (*Figure 6C*). Additionally, wortmannin treatment reduced the overall phosphorylation level of cytosolic HSCB, but not that of TACC3, in EPO-induced K562 cells and E 2-day HSCs (*Figure 6C*). We then identified the physical interaction between cytosolic HSCB and PIK3R1, which is the P85-alpha subunit of PI3K, via co-IP experiments (*Figure 6D*). These data indicated that HSCB phosphorylation by PI3K was indispensable for its functionalization by the EPO/EPOR signaling. In line with these findings, wortmannin treatment significantly inhibited the erythropoiesis of both K562 cells and CD34+CD90+HSCs (all p values were less than 0.01 or 0.001 in Student's *t* test, *Figure 6E–I*). Collectively, our results clarified that phosphorylation of HSCB by PI3K was necessary for its functionalization during human erythropoiesis. Since HSCB plays a central role during ISC delivery, we also investigated if HSCB phosphorylation is required for its proper functioning during ISC biogenesis. We analyzed the mitochondrial and cytosolic aconitase (m-aconitase and c-aconitase, respectively) activities of EPO-induced K562 cells and E 2-day HSCs and those treated wortmannin. As shown in *Figure 6—figure supplement 2*, wortmannin treatment did not significantly affect mitochondrial and cytosolic aconitase activities in EPO-induced K562 cells and E 2-day HSCs, suggesting that HSCB phosphorylation does not affect its ISC delivery function.

## Activation of the PI3K–HSCB axis was also necessary for the megakaryopoiesis of CD34+CD90+ HSCs

Given that PI3K is also activated in the TPO/MPL signaling and that FOG1 is also crucial for megakaryopoiesis, we finally tested whether the PI3K–HSCB signaling axis is necessary for the megakaryopoiesis of CD34+CD90+HSCs. As displayed in *Figure 7A–C*, the CD34+CD90+HSCs transfected with

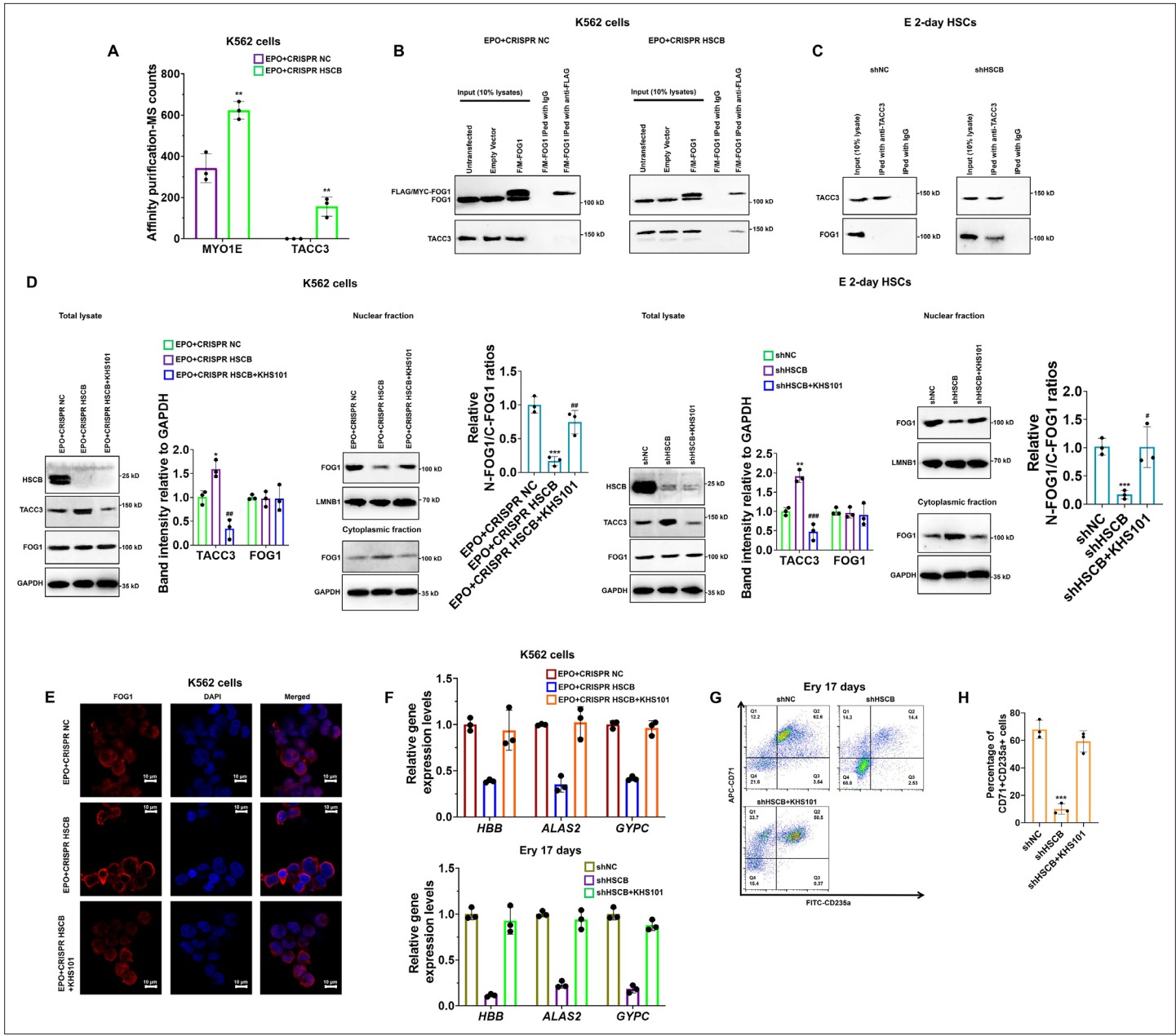

**Figure 4.** Deficiency of heat shock cognate B (HSCB) blocked FOG1 nuclear translocation by enhancing its interaction with transforming acidic coiled-coil containing protein 3 (TACC3). (**A**) A histogram showing the two differential binding proteins (DBPs) of FOG1 between the EPO+CRISPR NC and EPO+CRISPR HSCB groups. Error bars denote standard deviations; ** signifies p < 0.01 compared with the EPO+CRISPR NC group, two-sided Student's *t* test. Co-immunoprecipitation (co-IP) assays in (**B**) K562 cells and (**C**) E 2-day hematopoietic stem cells (HSCs) indicated that HSCB deficiency promoted the binding between FOG1 and TACC3. (**D**) Western blotting analyses on K562 cells and E 2-day HSCs and (**E**) immunofluorescence (IF) assays on K562 cells revealed that treatment with KHS101, a small-molecule inhibitor that could reduce the activity and protein abundance of TACC3, decreased TACC3 protein levels and increased relative nuclear FOG1 (N-FOG1) to cytoplasmic FOG1 (C-FOG1) ratios. Scale bars for IF assays indicate 10 μm. Error bars denote standard deviations; * and ** in panel (**D**), respectively, denote p values less than 0.05 and 0.01 compared with the EPO+CRISPR NC or shNC group, while #, ##, and ###, respectively, mean p < 0.05, p < 0.01, and p < 0.001 relative to the EPO+CRISPR HSCB or shHSCB group, two-sided Student's *t* test. (**F**) qRT-PCR analyses on HSCB-deficient erythropoietin (EPO)-induced K562 cells and HSCB-deficient Ery 17 days demonstrated recovered mRNA expression levels of erythroid-specific genes after KHS101 treatment. (**G**) Representative flow cytometry scatter plots and (**H**) statistics of three biologically independent flow cytometry assays showing that the percentages of CD71+CD235a+progenitors among HSCB-deficient Ery 17 days recovered after KHS101 treatment. Error bars denote standard deviations; *** means p < 0.001 compared with the shNC group, two-sided Student's *t* test.

The online version of this article includes the following source data and figure supplement(s) for figure 4:

*Figure 4 continued on next page*

*Figure 4 continued*

**Source data 1.** Labeled and raw Western blots for *Figure 4B–D*.

**Figure supplement 1.** Co-immunoprecipitation (co-IP) assays detected no interaction between FOG1 and MYO1E in the shNC and shHSCB groups of E 2-day hematopoietic stem cells (HSCs).

**Figure supplement 1—source data 1.** Labeled and raw Western blots for *Figure 4—figure supplement 1*.

the shHSCB plasmid (shHSCB group) and those treated with wortmannin (Wortmannin group) exhibited significantly reduced proportions of CD41a+CD42b+ cells and lower mRNA expression levels of megakaryocyte-specific genes *VWF* and *GP1BA* (all p values were less than 0.001 in Student's *t* test), while the megakaryopoiesis of CD34+CD90+HSCs transfected with the shHSCB plasmid and treated with KHS101 (shHSCB+KHS101) was undisrupted. Since most cells from the shHSCB group were CD34+CD38+CD45RA−IL3RA− (*Figure 7D*), an expression pattern most similar to that observed on CD34+CD90+HSCs induced for megakaryopoiesis for 3 days (MK 3-day HSCs) (*Figure 7—figure supplement 1*), MK 3-day HSCs were utilized for the following analyses. As exhibited in *Figure 7E*, the shHSCB and Wortmannin groups of MK 3-day HSCs had significantly higher levels of TACC3 and decreased relative N-FOG1/C-FOG1 ratios (all p values were less than 0.05 in Student's *t* test), while the shHSCB+KHS101 group possessed a lower level of TACC3 and a slightly increased level of FOG1 nuclear translocation. Additionally, wortmannin treatment reduced the overall phosphorylation level of HSCB and decreased its interaction with TACC3 (*Figure 7F*). We also determined the nuclear abundance of FOG1 on days 0, 3, 7, and 14 during CD34+CD90+HSCs megakaryopoiesis and found that the level of N-FOG1 peaked on day 3 and declined thereafter (*Figure 7G*). Taken together, these data were consistent with those observed in erythroid progenitors, indicating that the PI3K–HSCB signaling axis was also important for the normal megakaryopoiesis of CD34+CD90+HSCs.

## Discussion

FOG1 is a binding partner of GATA1 that exerts a vital function during erythropoiesis and megakaryopoiesis, as FOG1-deficient mice displayed megakaryopoiesis failure and impaired erythropoiesis (*Tsang et al., 1998*). A previous study suggested that FOG1 was needed for the formation of megakaryocyte/erythroid progenitors (MEPs), while GATA1 was required for the development of committed erythroid progenitors (*Mancini et al., 2012*). Given the biological significance of FOG1, it is crucial to understand how FOG1 is regulated by erythroid and megakaryocytic signals to ensure its timely activation or expression during erythropoiesis and megakaryopoiesis. Herein, we have established a PI3K–HSCB signaling axis that facilitated FOG1 nuclear translocation upon activation of the EPO/EPOR and TPO/MPL signaling pathways during human erythropoiesis and megakaryopoiesis (*Figure 7H*).

Although heme biosynthesis depends on an intact ISC assembly machinery, deficiencies in proteins of the ISC assembly complex, namely ISCU, NFS1, and LYRM4 have not been associated with erythropoietic consequences in both human subjects and cell models (*Ye and Rouault, 2010*). In contrast, however, deficiencies in proteins responsible for ISC transfer, such as GLRX5, HSPA9, and HSCB, have been tightly linked with sideroblastic anemia phenotypes (*Ducamp and Fleming, 2019*). This could be due to that ISC biogenesis deficiencies are tolerable in erythrocytes, while the proteins responsible for ISC transfer possess moonlighting functions during erythropoiesis. Our research was initially inspired by the findings that a CSA proband carrying heterozygous HSCB mutations developed moderate thrombocytopenia and mild neutropenia (*Crispin et al., 2020*). It is not surprising that mutations in proteins responsible for ISC transfer can cause anemia because two of the eight enzymes catalyzing heme biosynthesis, namely aminolevulinate dehydratase and ferrochelatase, were identified to be iron–sulfur proteins (*Dailey et al., 1994*; *Liu et al., 2020*). However, it was not clear why the HSCB-deficient proband also exhibited moderately reduced numbers of platelets. Therefore, we tested whether HSCB is implicated with erythropoiesis and megakaryopoiesis of human CD34+CD90+HSCs. Our results indicated that HSCB knockdown detained the erythropoiesis and megakaryopoiesis of HSCs at early stages because most of the resultant progenitors were CD34+CD38+CD45RA−IL3RA−CD71−CD235a− or CD34+CD38+CD45RA−IL3RA−CD41a−CD42b−, corresponding to common myeloid progenitors or MEPs as per previous studies (*Edvardsson et al., 2006*; *Kwon et al., 2021*). Further analysis of the underlying mechanisms revealed that HSCB deficiency mainly impaired FOG1

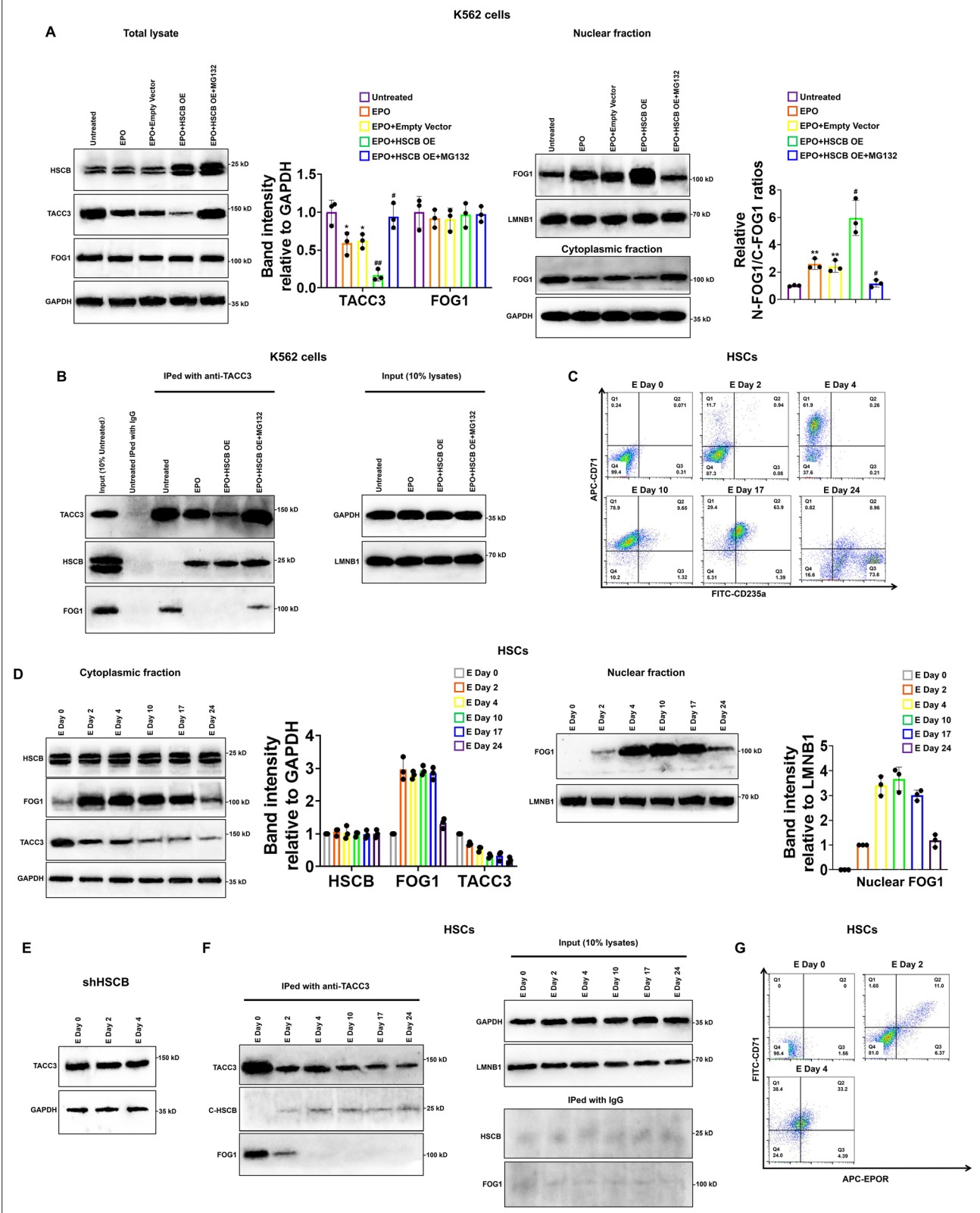

**Figure 5.** Heat shock cognate B (HSCB) bound with and mediated the proteasomal degradation of TACC3 to facilitate FOG1 nuclear translocation upon erythropoietin (EPO)/erythropoietin receptor (EPOR) signaling activation. (**A**) Western blotting analyses indicated that EPO treatment decreased the level of TACC3 and increased the relative N-FOG1/C-FOG1 ratio, HSCB overexpression augmented the effects of EPO treatment on the protein level of TACC3 and FOG1 nuclear translocation, whereas treatment with the proteasome inhibitor MG132 abrogated the effects of EPO treatment +

*Figure 5 continued on next page*

*Figure 5 continued*

HSCB overexpression on the protein level of TACC3 and FOG1 nuclear translocation. Error bars denote standard deviations; * and **, respectively, represent p values less than 0.05 and 0.01 compared with the Untreated group, while # and ##, respectively, mean p < 0.05 and p < 0.01 relative to the EPO+Empty Vector group, two-sided Student's *t* test. (**B**) Co-immunoprecipitation (co-IP) results showing that TACC3 interacted with HSCB when the K562 cells were treated with EPO, whereas it interacted with FOG1 in the untreated and EPO+HSCB OE + MG132 K562 cells. (**C**) Flow cytometry scatter plots exhibiting the levels of CD71 and CD235a on the cell surface of CD34+CD90+HSCs induced for erythropoiesis for 0, 2, 4, 10, 17, and 24 days. The flow cytometry analyses were repeated twice with similar findings to ensure reproducibility. (**D**) Western blotting analyses of the levels of cytoplasmic FOG1, TACC3, and nuclear FOG1 in CD34+CD90+HSCs induced for erythropoiesis for 0, 2, 4, 10, 17, and 24 days. (**E**) Western blotting analyses of the level of TACC3 in HSCB-deficient HSCs induced for erythropoiesis for 0, 2, and 4 days. (**F**) Co-IP analyses of the binding affinities of TACC3 with HSCB and FOG1 in CD34+CD90+HSCs induced for erythropoiesis for 0, 2, 4, 10, 17, and 24 days. (**G**) Flow cytometry scatter plots showing the levels of CD71 and EPOR on the cell surface of CD34+CD90+HSCs induced for erythropoiesis for 0, 2, and 4 days. The Western blotting, co-IP, and flow cytometry data highlighted that the EPO/EPOR signaling was indispensable for the functionalization of HSCB in CD34+CD90+HSCs induced for erythropoiesis.

The online version of this article includes the following source data for figure 5:

**Source data 1.** Labeled and raw Western blots for *Figure 5A, B, D–F*.

---

nuclear translocation. The only known function of HSCB is to deliver nascent ISCs to recipient proteins. Interestingly, knockdown of proteins responsible for ISC assembly did not significantly affect erythropoiesis of HSCs, thus highlighting an ISC delivery independent function of HSCB during erythropoiesis and megakaryopoiesis. HSCB is a chaperone expressed in both mitochondria and cytosol. In this study, we were able to detect two bands for HSCB with WB, among which the upper and lower bands, respectively, represent cytosolic and mitochondrial HSCB (*Liu et al., 2020*). As revealed by *Figure 5D* and *Figure 6A*, the subcellular distribution of HSCB was neither significantly altered during erythropoiesis of CD34+CD90+HSCs nor significantly affected by ruxolitinib and wortmannin treatments in K562 cells. Therefore, it seems that the subcellular localization of HSCB does not change during erythropoiesis or correlate with its phosphorylation status. However, whether the phosphorylation status of HSCB affects its ISC delivery function warrants further investigation. Additionally, since HSCB can bind with a wide array of intracellular proteins (*Maio et al., 2017*), we speculate that the protein may be involved in many biological processes other than nascent ISC delivery. In this context, we hope our study may inspire more research efforts on ISC delivery independent functions of HSCB.

TACC3 belongs to a family of TACC-domain-containing proteins associated with mitotic spindles and centrosomes (*Ding et al., 2017*). The protein participates in cell division by regulating the organization of spindle and microtubule nucleation during mitosis (*Ding et al., 2017*). Recently, TACC3 has attracted increasing research interest because it was found aberrantly expressed in cancers (*Ding et al., 2017*; *Saatci et al., 2023*). In addition to its association with tumorigenesis, TACC3 has also been implicated with erythropoiesis. According to a previous study, TACC3 could inhibit FOG1 nuclear translocation to suppress the erythropoiesis of mouse MEL and G1ER cells (*Garriga-Canut and Orkin, 2004*). Herein we found that TACC3 exerted a similar function in human erythroid and megakaryocytic progenitor cells because, in the absence of HSCB, the protein could bind with and prevent the nuclear translocation of FOG1. Then we observed that treatment with KHS101, a small-molecule inhibitor that could decrease the activity and protein abundance of TACC3, alleviated the inhibitory effect of TACC3 on erythropoiesis upon HSCB deficiency. These data clarified the mechanism through which HSCB deficiency inhibited FOG1 nuclear translocation and consequently impaired erythropoiesis and megakaryopoiesis. In sight of the functions of TACC3 during cell division and erythropoiesis/megakaryopoiesis, it is possible that TACC3 plays a crucial role in maintaining the balance between cell proliferation and differentiation, which of course warrants further investigation. Furthermore, given that KHS101 could decrease TACC3 protein level to facilitate FOG1 nuclear translocation, this small-molecule reagent may have the potential to treat diseases caused by reduced activity or protein level of FOG1.

The precise mechanisms through which EPO/EPOR and TPO/MPL signaling pathways regulate lineage-specific transcription events during erythropoiesis and megakaryopoiesis remain incompletely understood. In a previous research, expression of a constitutively activated Stat5a mutant in $Jak2^{-/-}$ and $EpoR^{-/-}$ mouse fetal liver cells was found to rescue their proliferation defects independently of EPO (*Grebien et al., 2008*). Additionally, the authors also observed that transplantation of $Jak2^{-/-}$ fetal liver cells expressing the Stat5a mutant into irradiated mice was able to maintain erythropoiesis and myelopoiesis of the mice (*Grebien et al., 2008*). These data verified the important function

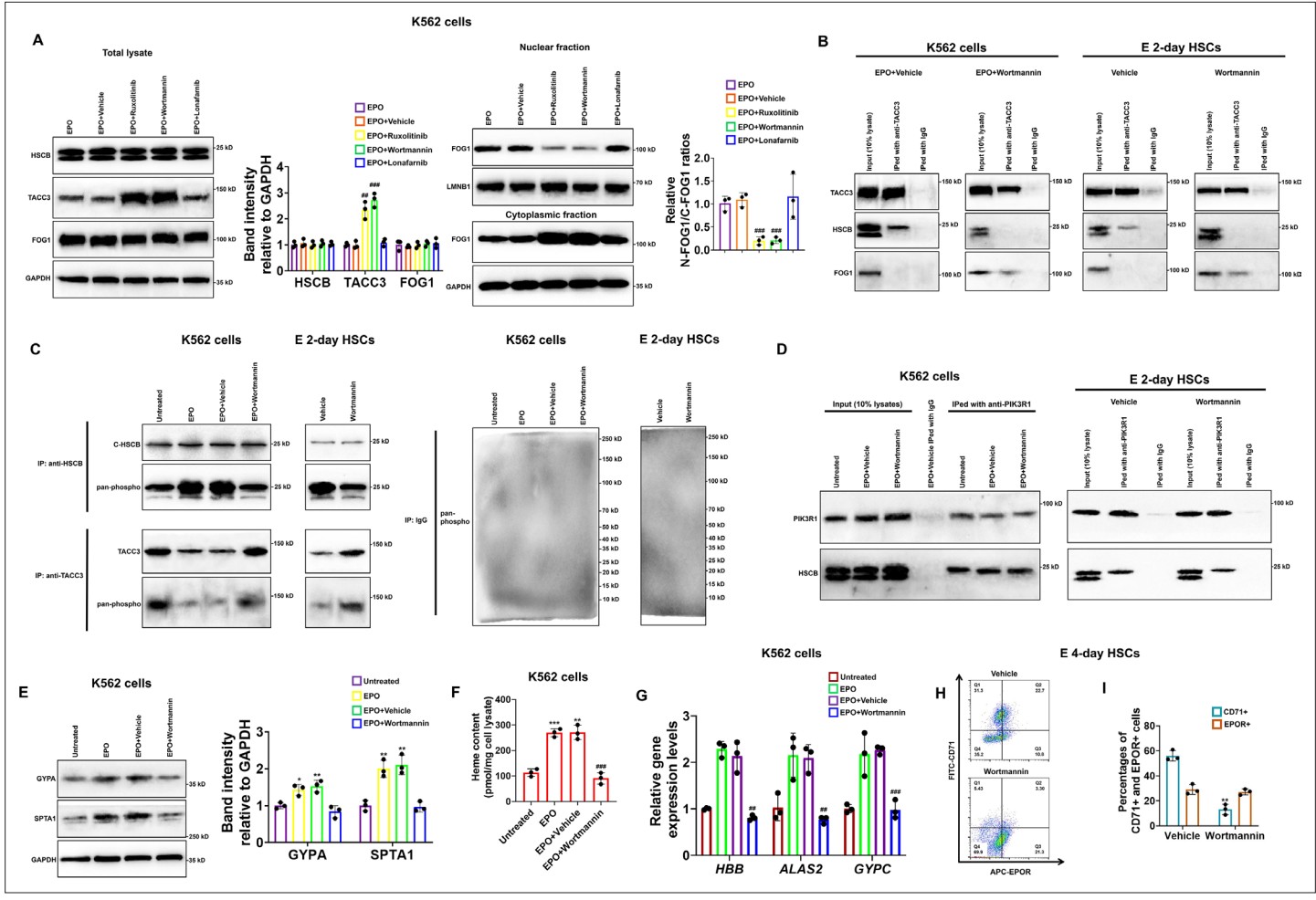

**Figure 6.** Heat shock cognate B (HSCB) phosphorylation by phosphoinositol-3-kinase (PI3K) was necessary for its functionalization during erythropoiesis of K562 cells and CD34+CD90+HSCs. (**A**) Western blotting analyses demonstrated that treatment with ruxolitinib and wortmannin, which are, respectively, inhibitors against Janus tyrosine kinase 2 (JAK2) and PI3K, increased the protein level of TACC3 and decreased FOG1 nuclear translocation. Error bars denote standard deviations; #, ## , and ###, respectively, mean p < 0.05, p < 0.01, and p < 0.001 relative to the EPO+Vehicle group, two-sided Student's t test. (**B, C**) Co-immunoprecipitation (co-IP) assays reflected that wortmannin treatment impaired the interaction between TACC3 and HSCB, promoted the binding between TACC3 and FOG1, and reduced the phosphorylation level of cytosolic HSCB in erythropoietin (EPO)-induced K562 cells and E 2-day hematopoietic stem cells (HSCs). (**D**) Co-IP experiments verified the physical interaction between cytosolic HSCB and the P85-alpha subunit of PI3K (PIK3R1). (**E**) Western blotting, (**F**) heme content, and (**G**) qRT-PCR assays confirmed that wortmannin treatment inhibited the erythroid differentiation of EPO-induced K562 cells. Error bars denote standard deviations; *, **, and ***, respectively, denote p values less than 0.05, 0.01, and 0.001 relative to the Untreated group, while ## and ###, respectively, represent p values less than 0.01 and 0.001 compared with the EPO+Vehicle group, two-sided Student's t test. (**H**) Representative flow cytometry scatter plots and (**I**) statistics of three biologically independent flow cytometry assays exhibiting significantly reduced percentages of CD71+ cells among E 4-day HSCs in the Wortmannin group. Error bars denote standard deviations; ** means p < 0.01 compared with the Vehicle group, two-sided Student's t test.

The online version of this article includes the following source data and figure supplement(s) for figure 6:

**Source data 1.** Labeled and raw Western blots for *Figure 6A–E*.

**Figure supplement 1.** Western blotting data verified the effects of ruxolitinib, wortmannin, and lonafarnib treatments on erythropoietin (EPO)-induced K562 cells.

**Figure supplement 1—source data 1.** Labeled and raw Western blots for *Figure 6—figure supplement 1*.

**Figure supplement 2.** Mitochondrial and cytosolic aconitase activities of Untreated, EPO, EPO+Vehicle, and EPO+Wortmannin groups of K562 cells as well as Vehicle and Wortmannin groups of E 2-day hematopoietic stem cells (HSCs).

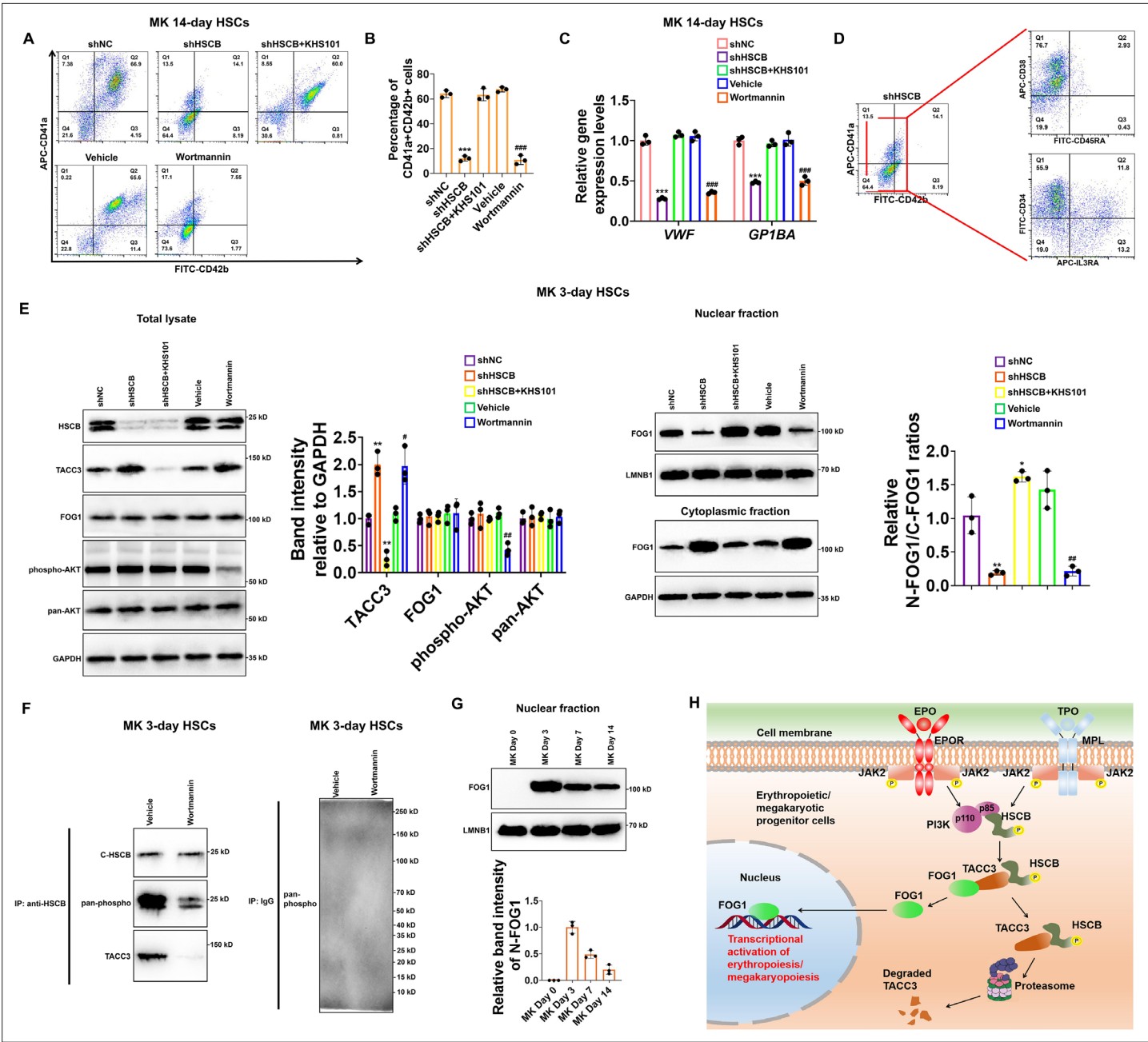

**Figure 7.** Megakaryopoiesis of CD34+CD90+HSCs also required the activation of the phosphoinositol-3-kinase (PI3K)–heat shock cognate B (HSCB) signaling axis. (**A**) Representative flow cytometry scatter plots and (**B**) statistics of three biologically independent flow cytometry assays revealing significantly reduced percentages of CD41a+CD42b+ cells among the MK 14-day hematopoietic stem cells (HSCs) in the shHSC and Wortmannin groups. (**C**) qRT-PCR assays indicated decreased mRNA expression levels of megakaryocyte-specific genes *VWF* and *GP1BA* in the MK 14-day HSCs of the shHSC and Wortmannin groups. Error bars denote standard deviations; *** and ###, respectively, denote p < 0.001 relative to the shNC and Vehicle groups, two-sided Student's *t* test. (**D**) Flow cytometry scatter plots showing that the majority of cells from the shHSCB group of MK 14-day HSCs were CD34+CD38+CD45RA−IL3RA−. (**E**) Western blotting analyses demonstrated that both knocking down HSCB and wortmannin treatment could increase the protein level of TACC3 while reduce FOG1 nuclear translocation, whereas treating HSCB-deficient HSCs with KHS101 decreased the protein level of TACC3 while enhanced FOG1 nuclear translocation in MK 3-day HSCs. Error bars denote standard deviations; *, ** and ***, respectively, represent p values less than 0.05, 0.01 and 0.001 compared with the shNC group, while # and ##, respectively, mean p < 0.05 and p < 0.01 relative to the Vehicle group, two-sided Student's *t* test. (**F**) Co-immunoprecipitation (co-IP) analyses revealed decreased phosphorylation level of cytosolic HSCB and impaired interaction between HSCB and TACC3 in MK 3-day HSCs treated with wortmannin. (**G**) Western blotting analyses of nuclear FOG1 levels in CD34+CD90+HSCs induced for megakaryopoiesis for 0, 3, 7, and 14 days. (**H**) A schematic diagram summarizing the main findings of this study.

The online version of this article includes the following source data and figure supplement(s) for figure 7:

*Figure 7 continued on next page*

*Figure 7 continued*

**Source data 1.** Labeled and raw Western blots for *Figure 7E–G*.

**Figure supplement 1.** Flow cytometry scatter plots showing that the majority of MK 3-day hematopoietic stem cells (HSCs) were CD34+CD38+CD45RA−IL3RA−.

of Stat5a as a downstream effector of the Epo/EpoR signaling pathway during murine erythropoiesis and myelopoiesis. However, since STAT5 possesses a variety of target genes, why the protein is specifically required for erythropoiesis and myelopoiesis warrants further investigation. Another early study demonstrated that wortmannin could impede EPO-induced K562 cell erythropoiesis but did not inhibit erythropoiesis of the cells triggered by hemin (*Kubota et al., 1996*). Since wortmannin is an inhibitor for PI3K, this study revealed that PI3K plays a large part in EPO-induced erythropoiesis of K562 cells. Later, it was demonstrated that the PI3K–AKT signaling cascade could directly activate GATA1 to stimulate erythropoiesis (*Zhao et al., 2006*). In this study, we confirmed the importance of PI3K activity during erythropoiesis and megakaryopoiesis and established a novel molecular mechanism. Our data indicated that PI3K could bind with and phosphorylate HSCB to promote its interaction with TACC3 to induce TACC3 degradation, thereby facilitating FOG1 nuclear translocation during erythropoiesis of K562 human erythroleukemia cells and HSCs, as well as during megakaryopoiesis of HSCs. Interestingly, the nuclear and cytoplasmic GATA1 expression levels were not regulated by this PI3K–HSCB axis, as revealed by the data presented in *Figure 2F* and *Figure 3A*, as well as the TMT-MS counts provided in *Supplementary file 6*. These new findings explain how EPO/EPOR and TPO/MPL signaling pathways simultaneously activate GATA1 and FOG1 to ensure that this duo can function in a concerted and timely manner during human erythropoiesis and megakaryopoiesis. In addition to the EPO/EPOR and TPO/MPL signaling pathways, PI3K is also activated by the SCF–c-KIT pathway (*Lennartsson and Rönnstrand, 2012*). Since the c-KIT and MRL receptors are expressed early on HSCs (*Lennartsson and Rönnstrand, 2012*), it is conceivable that the timely activation of GATA1 and FOG1 during erythropoiesis and megakaryopoiesis also requires the participation of other differentiation stage-specific mechanisms regulating the abundance of these proteins. This idea can be supported by our findings on HSCs induced for erythropoiesis for different periods of time. As shown in *Figure 5D*, while the levels of cytoplasmic TACC3 and FOG1 exhibited no significant changes from days 2 to 4 of erythropoiesis, the level of N-FOG1 increased by more than three folds during the same period, suggesting that the functionalization of FOG1 is determined not only by the activation of the PI3K–HSCB axis, but also by the overall expression level of this protein. Additionally, although we failed to detect nuclear FOG1 in uninduced CD34+CD90+HSCs (*Figure 5D*), we did detect a faint band of FOG1 in the cytoplasmic fraction. Given that HSCB was already expressed in uninduced CD34+CD90+HSCs, why the SCF–c-KIT–PI3K pathway failed to promote FOG1 nuclear translocation in these cells warrants further investigation.

In addition to the erythroid and megakaryocytic lineages, the interaction between FOG1 and GATA1 can also regulate the development of mast cells. According to a previous study, GATA1 could exert its functions in a FOG-1-independent manner in the mast cell lineage (*Cantor et al., 2008*). Furthermore, FOG1 was found to impede the commitment of HSCs to the mast cell lineage, and its overexpression could redirect committed mast cell progenitors into granulocytic, erythroid, and megakaryocytic lineages (*Cantor et al., 2008*). A subsequent study delved into the mechanisms by which FOG1 inhibits mast cell lineage development by utilizing a GATA1 mutant that could not bind with FOG1 (*Chlon et al., 2012*). The study revealed that FOG1 could affect the chromatin occupancy of GATA1 to regulate the expression of lineage-specific genes (*Chlon et al., 2012*). Since our findings demonstrated that inhibition of cytosolic HSCB phosphorylation could retain FOG1 in the cytoplasm, it will be interesting to investigate whether these GATA1-expressing myeloid progenitors in which FOG1 is kept cytoplasmic could be driven toward the mast cell fate. Additionally, whether and how HSCB is implicated with the mast cell lineage-specifying events warrant further investigation.

This study has some limitations that should be acknowledged. First, we utilized the K562 cell line as one of the cell models in this study because it has the potential to differentiate into erythroid, megakaryocyte, and macrophage lineages and thus represents a useful tool in studying the molecular mechanisms underlying early-stage erythropoiesis and megakaryopoiesis (*Nandakumar et al., 2016*; *Sutherland et al., 1986*). However, although K562 cells have been widely utilized to study erythropoiesis (*Andersson et al., 1979*; *Cao et al., 2020*), and the capability of EPO to induce K562

erythropoiesis has been demonstrated by this study and previous research (*Hoffman et al., 1979*; *Kubota et al., 1996*), this human erythroleukemia cell line might not faithfully represent the normal physiological processes of erythropoiesis. Fortunately, all our major conclusions can be successfully recapitulated in primary cells. A second limitation is that we did not focus on LYAR in this study because the difference in nuclear abundance of this protein between the shNC and shHSCB groups of E 2-day HSCs did not reach a statistically significant level (*Figure 2F–G*). However, given the established function of LYAR during late erythropoiesis, it will be intriguing to explore whether this protein is also implicated with HSCB functionalization during human erythropoiesis. A third limitation is that we did not clarify the detailed mechanisms through which HSCB promotes TACC3 degradation, which warrant further exploration. Finally, since we did not have access to biological samples from the patient harboring the *HSCB* mutation and that HSCB conditional knockout mice had almost no erythroid cells in their bone marrow, we were not able to further validate our findings in in vivo models.

In spite of these limitations, our study established a PI3K–HSCB axis that acted downstream of the EPO/EPOR and TPO/MPL signaling pathways to induce proteasomal degradation of TACC3, which otherwise detained FOG1 in the cytoplasm, in K562 cells and HSCs. This work elucidates an essential iron–sulfur cluster delivery independent function of HSCB in translating differentiation signals into downstream transcriptional events during human erythropoiesis and megakaryopoiesis.

# Materials and methods
## Culturing and treatment of K562 cell line and HSCs

Human K562 erythroleukemia cells were purchased from Procell Life Science&Technology Co, Ltd (China). These cells have been authenticated by STR profiling and tested negative for mycoplasma contamination. The cells were maintained in Iscove's modified Dulbecco medium (IMDM, PM150510, Procell, China) supplemented with 10% FBS (085-150, Wisent Bioproducts, Canada), 100 U/ml penicillin and 100 μg/ml streptomycin (PB180120, Procell, China) in a 37°C, 5% $CO_2$ incubator. For induction of erythropoiesis, K562 cells were cultured in IMDM without FBS (FBS deprivation treatment) for 24 hr and then treated with 5 U/ml human recombinant EPO (GP20193, GlpBio, USA) for 48 hr. After the FBS deprivation treatment, K562 cells were incubated for 48 hr with 300 nM ruxolitinib (GC14191), 200 nM wortmannin (GC12338), 2 μM lonafarnib (GC10330) dissolved in dimethylsulfoxide (DMSO) and 10 μM MG132 (GC10383, all from GlpBio, USA) dissolved in phosphate-buffered saline (PBS) for inhibiting the activities of JAK2, PI3K, RAS, and proteasome, respectively, and with only DMSO for the 'vehicle' treatment. To reduce the protein level of transforming acidic coiled-coil containing protein 3 (TACC3), K562 cells were treated with 5 μM KHS101 (GC13582, GlpBio, USA) dissolved in PBS.

Cord-blood-derived CD34+human HSCs were originally obtained from healthy female donors aged between 24 and 27 years and were commercially available at Precision BioMedicals Co, Ltd, China (903410). The CD34+HSCs population was first expanded by incubating the cells in StemSpan Serum-Free Expansion Medium (SFEM) II (09655) plus StemSpan CD34+Expansion Supplement that contains recombinant human fms-like tyrosine kinase 3 ligand, SCF, IL3, IL6 and TPO (02691, all from STEM-CELL Technologies, Canada) for 7 days in a 37°C, 5% $CO_2$ incubator. After that, CD34+CD90+HSCs were purified by cell sorting (described below). The HSCs were cultured for 2, 4, 10, 17, or 24 days in StemSpan SFEM II plus StemSpan Erythroid Expansion Supplement that contains recombinant human SCF, IL3, and EPO (02692, STEMCELL Technologies, Canada) for induction of erythropoiesis, or for 3 or 14 days in StemSpan SFEM II plus StemSpan Megakaryocyte Expansion Supplement that contains SCF, IL-6, IL-9, and TPO (02696, STEMCELL Technologies, Canada) for induction of megakaryopoiesis. The concentration of the HSCs or the subsequent erythroid/megakaryocytic progenitors was maintained below $1 \times 10^5$ cells/ml during all the aforementioned experiments. The levels of PI3K activity and TACC3 protein expression were reduced by culturing the cells with 100 nM wortmannin (GC12338) and 4 μM KHS101 (GC13582, all from GlpBio, USA), respectively.

## Gene knockdown, overexpression, and knockout

Plasmids carrying shRNAs against human *ISCU* (sc-270108-SH), *NFS1* (sc-75911-SH), and *HSCB* (sc-75306-SH) genes, as well as that carrying a negative control (NC) shRNA (sc-108060), were obtained from Santa Cruz Biotechnology, China. Plasmids overexpressing human FOG1 (pcDNA3.1-FOG1 OE), FLAG/MYC (F/M)-tagged FOG1 (pCMV-F/M-FOG1 OE), and human HSCB (pcDNA3.1-HSCB OE), and

the corresponding empty vectors were synthesized and provided by Fenghui Biotechnology Co, Ltd, China. The HSCB CRISPR/Cas9 KO Plasmid (h) (sc-409377), HSCB HDR Plasmid (h) (sc-409377-HDR), and Control CRISPR/Cas9 Plasmid (sc-418922) were procured from Santa Cruz Biotechnology, China. These plasmids were transfected into CD34+CD90+HSCs by utilizing the RFect$^{SP}$ Plasmid DNA Transfection Reagent (21027, Biodai, China). After transfection with shRNA plasmids, the HSCs were first cultured in StemSpan Serum-Free Expansion Medium (SFEM) II (09655, STEMCELL Technologies, Canada) supplemented with 2 µg/ml puromycin (GC16384, GlpBio, USA) for 72 hr and then induced for erythropoiesis or megakaryopoiesis. Stable transfection of shRNA plasmids was maintained by culturing the cells with 4 µg/ml puromycin. For FOG1 overexpression, CD34+CD90+HSCs were transfected with a pcDNA3.1-FOG1-OE plasmid, cultured in StemSpan SFEM II for 48 hr and then induced for erythropoiesis. During the erythropoiesis process, the cells were transfected with the pcDNA3.1-FOG1-OE plasmid once every 72 hr (see *Figure 1—figure supplement 1* for a schematic diagram).

For FOG1, FLAG/MYC-FOG1, and HSCB overexpression in K562 cells, the cells were transfected with the corresponding plasmids, cultured for 48 hr, and harvested for analyses. For silencing of HSC expression, K562 cells were co-transfected with an HSCB CRISPR/Cas9 KO Plasmid and an HSCB HDR Plasmid with the RFect$^{SP}$ Plasmid DNA Transfection Reagent. Following that, stably transfected K562 cells were selected for 7 days by 5 µg/ml puromycin and then subjected to different treatments.

## Enzyme activity assays

The activities of aconitase and PDH were, respectively, analyzed with the Aconitase Activity Assay Kit (ab109712, Abcam, UK) and the PDH Enzyme Activity Microplate Assay Kit (ab109902, Abcam, UK). For aconitase activity assays, $1 \times 10^6$ Ery 17 days were collected by centrifugation at $1500 \times g$ for 5 min, washed with PBS for two times, and then resuspended in the Aconitase preservation solution provided with the kit. The cell suspension was incubated in the presence of the detergent provided with the kit on ice for 30 min and then centrifuged at $20,000 \times g$ for 10 min. The supernatant was diluted into a concentration of 25 µg/50 µl with the Buffer provided with the kit, mixed with 200 µl of Assay buffer, and measured for optical density at the wavelengh of 240 nm during a 30-min interval under room temperature. The whole-cell-lysate aconitase activity was calculated as per the detailed instructions provided in the protocol of the kit. For PDH activity assays, $2 \times 10^6$ cells were collected and washed as described above. The lysates were prepared with the Detergent solution, diluted into a concentration of 25 µg/50 µl with a prepared 1× Buffer, and measured for optical density at the wavelengh of 450 nm during a 30-min interval under room temperature. The PDH activity was calculated as per the detailed instructions provided in the protocol. All these colorimetric assays were carried out based on a Tecan Infinite M Nano microplate reader (Switzerland).

## Heme content assay

Intracellular heme contents were determined with a method described previously (*Sassa, 1976*). Briefly, whole-cell lysates were prepared with the RIPA Lysis Buffer (PC101, Epizyme, China). Afterwards, the lystates were analyzed with the Omni-Easy Instant BCA Protein Assay Kit (ZJ102, Epizyme, China) for protein concentration determination. Then 30 µg protein samples were added into 500 µl of 2 M oxalic acid, mixed thoroughly, and divided into two aliquots. One aliquot was heated at 100°C for 30 min, while the other was not heated (to correct for endogenous porphyrins). After cooling, fluorescence was determined at excitation and emission wavelengths of 400 and 662 nm, respectively, using the Tecan microplate reader. A standard curve was established based on hemin solutions to allow absolute quantification. The data are expressed as pmol heme per milligram (mg) of cell lysate.

## Flow cytometry

Briefly, $2 \times 10^5$ cells were rinsed and evenly dispersed in ice-cold PBS containing 10% FBS, incubated at 4°C for 1 hr in dark with fluorescein isothiocyanate- or allophycocyanin-conjugated primary antibodies, washed three times with the ice-cold PBS containing 10% FBS, and analyzed immediately with a CytoFLEX flow cytometer (Beckman Coulter, USA). The data were analyzed with Treestar FlowJo v.10.6.2. The gating strategies for cell populations were formulated based on flow cytometry findings with single stain controls for each sample (please see *Figure 1—figure supplement 2D* for an example). Information on the primary antibodies utilized for flow cytometry analyses is summarized in *Supplementary file 1—table 1*.

## Cell sorting

Positive selection of CD34+CD90+HSCs and negative selection of CD71−CD235a− progenitors were, respectively, carried out with the CD34 MicroBead Kit UltraPure (130-100-453) plus CD90 MicroBeads, human (130-096-253) and CD71 MicroBeads, human (130-046-201) plus CD235a (Glycophorin A) MicroBeads, human (130-050-501) combinations based on the Mini & MidiMACS Starting Kit (130-091-632) (all from Miltenyi Biotec, Germany) in accordance with the manufacturer's protocols.

## Quantitative real-time PCR

Total RNA was isolated from erythroid/megakaryocytic progenitor or K562 cells using the RNA Isolation Kit (Omn-02, Omiget, China) and reverse-transcribed into cDNA with the All-In-One 5X RT MasterMix (G592, ABM, Canada). For qRT-PCR, the 20 μl reaction systems were established with the cDNA samples and the gene-specific primers listed in *Supplementary file 1—table 2* based on the BlasTaq 2X qPCR MasterMix (G891, ABM, Canada). All reactions were performed on the QuantStudio 3 Real-Time PCR System (Applied Biosystems, USA). Human *GAPDH* gene served as the internal reference gene, relative to which the mRNA expression levels of our target genes were calculated with the $2^{-\Delta\Delta Ct}$ algorithm. Four replicates were set up for each sample, and three biologically independent samples were analyzed.

## Immunofluorescence assay

K562 cells were washed with PBS, incubated in the 4% Paraformaldehyde Fix Solution (P0099, Beyotime, China) for 20 min, and washed thrice and resuspended with the Immunol Staining Wash Buffer (P0106, Beyotime, China). The cell suspension was dropped onto a glass slide and allowed to dry for 30 min. Then the slide was sequentially incubated in the Immunol Staining Blocking Buffer (P0102, Beyotime, China) for 60 min under ambient temperature and then in a ZFPM1 Rabbit pAb solution (1:100 diluted in the wash buffer, A18183, ABclonal, China). Afterwards, the slide was washed three times with the wash buffer and stained sequentially with the Immunol Fluorence Staining Kit with Alexa Fluor 647-Labeled Goat Anti-Rabbit IgG (P0180, Beyotime, China) and 4',6-diamidino-2-phenylindole (DAPI, MBD0015, Sigma-Aldrich, USA) as per the manufacturer's instructions. Finally, the slide was observed with a Carl Zeiss LSM 880 AxioObserver Confocal Laser Scanning Microscope equipped with an Airyscan detector under room temperature (approximately 23°C) (Zeiss, Germany). The objective was Zeiss Plan-Apochromat ×63/1.40 Oil DIC M27 with a numerical aperture of 1.4. The imaging medium was ZEISS Immersol 518 F fluorescence free oil. The excitation wavelengths for Alexa Fluor 647 and DAPI were, respectively, 633 and 405 nm. The images were acquired and analyzed with the ZEN 2.6 (blue edition) software. For data processinig, we only added scale bars (10 μm) to the original images using the ZEN 2.6 (blue edition) software.

## Separation of cytosolic and mitochondrial fractions

Cytosolic and mitochondrial fractions of K562 cells were obtained using the Mitochondria Isolation Kit (89874, Thermo Scientific, USA).

## Co-IP analysis

Physical interactions between proteins and the phosphorylation status of target proteins were analyzed by the Classic Magnetic Protein A/G IP/Co-IP Kit (YJ201, Epizyme, China) in conjunction with WB analysis. The inputs used for WB in co-IP assays were 10% whole-cell lysates. For the negative control, bait proteins were captured with Normal Rabbit IgG (2729, Cell Signalling Technology, USA) or Mouse Control IgG (AC011, ABclonal, China). Information on the primary antibodies used to capture bait proteins for the co-IP assays is available in *Supplementary file 1—table 1*.

## Nuclear proteomics analysis

Nuclear extracts were prepared with the NE-PER Nuclear and Cytoplasmic Extraction Kit (78833, Thermo Scientific, USA) following the protocol provided by the manufacturer. For nuclear proteomics analysis, 500 μg of nuclear extracts were subjected to TMT-MS performed by CapitalBio Technology, China to characterize nuclear proteomic changes. Nuclear extracts of equal protein amounts were first subjected to proteolysis with trypsin (V5113, Promega) at 37°C for 16 hr. The resulting peptide fragments were then desalted with C18 Cartridge columns (WAT023590, Waters), freeze-dried, and

reconstituted in a 40 µl 0.1% formic acid solution. Equal amounts of peptide fragments from each experimental group were then labeled utilizing the TMTpro 16plex Isobaric Label Reagent (A44520, Thermo Fisher) as per the manufacturer's instructions. The labeled fragments were fractionated through high-performance liquid chromatography with the C18 Cartridge columns and analyzed by liquid chromatography–tandem mass spectrometry (LC–MS/MS) assays using a Q Exactive HF-X mass spectrometer (Thermo Fisher). Each assay involved a full-spectrum scan (Resolution 120,000, scan range 350–1500 $m/z$, AGC target 3e6, maximum IT 50 ms) and target MS/MS scans (Resolution 45,000, AGC target 5e4, maximum IT 80ms). The raw MS data were searched against the Uniprot database (uniprot-UP000005640_9606.fasta) for annotation with Proteome Discoverer Version 2.4 (Thermo Fisher) with the following parameters: Enzyme, trypsin; Max missed cleavages, 2; Fixed modifications, carbamidomethyl (C), TMTpro (N-term), and TMTpro (K); Variable modifications, oxidation (M) and acetyl (protein N-term); Peptide mass tolerance, ±10 ppm; Fragment mass tolerance, 0.02 Da; Peptide confidence, high. A peptide false discovery rate of ≤0.01 was applied as the threshold for filtering the identified peptides based on their precursor intensity fraction. The final MS counts for each protein in different experimental groups were compared using unpaired Student's $t$ tests. Nuclear proteins with a |log$_2$(fold change)| value of ≥0.415 and an unpaired $t$ test p value of <0.05 were deemed DENPs.

## Affinity purification-MS analysis

To explore the proteins exhibiting significantly different binding affinities with FOG1, the binding partners of F/M-tagged FOG1 in the EPO+CRISPR NC and EPO+CRISPR HSCB groups of K562 cells were purified by the Anti-FLAG M2 Magnetic Beads (M8823, Sigma-Aldrich, USA). Subsequently, these binding partners were subjected to TMT-MS, and their counts in the analysis were compared using unpaired $t$ test. A p value of <0.05 was considered to indicate proteins exhibiting significantly different binding affinities with FOG1 between the two groups of K562 cells (DBPs).

## Western blotting

Whole-cell lysates or lysates of cytosolic and mitochondrial fractions were prepared with the RIPA Lysis Buffer (PC101, Epizyme, China). Afterwards, the lysates were subjected to bicinchoninic acid assay (BCA) utilizing the Omni-Easy Instant BCA Protein Assay Kit (ZJ102, Epizyme, China) for determination of protein concentration. Equal amounts of protein samples (20–50 µg) were then separated by sodium dodecyl sulfate–polyacrylamide gel electrophoresis gels prepared with the PAGE Gel Fast Preparation Kit (PG112, Epizyme, China) and blotted onto 0.22 µm polyvinylidene fluoride membranes. After electrotransfer, the membranes were first incubated in 5% skimmed milk dissolved in Tris-buffered saline with Tween-20 (TBST) for 1 hr at room temperature, then in diluted primary antibodies (see *Supplementary file 1—table 1* for details) for 2–3 hr at room temperature or overnight at 4°C, and finally in appropriate 1:5000 diluted horseradish peroxidase-conjugated secondary antibodies (LF101 or LF102, Epizyme, China). Target protein bands were finally visualized using the Tanon High-sig ECL Western Blotting Substrate (180-501, Tanon, China). For band intensity analysis, ImageJ v. 1.53 was employed.

## Statistical analysis

Statistical data obtained in this study were analyzed by two-sided unpaired Student's $t$ test with GraphPad Prism 9.3 (GraphPad Software, USA). All data represent mean ± standard deviation of three independent biological replicates. p values less than 0.05, 0.01, and 0.001, respectively, signify significant (* or #), very significant (** or ##), and extremely significant (*** or ###) differences.

## Acknowledgements

This work was financially supported by the Jilin Province Science and Technology Development Plan Project (Grant No. 20220508072RC) and the High-level Talent Recruitment Project of Northeast Normal University for GL, as well as the Fundamental Research Funds for the Central Universities for GL (135113022) and Xiumei Jiang (135113009).

## Additional information

### Funding

| Funder | Grant reference number | Author |
|---|---|---|
| Jilin Provincial Scientific and Technological Development Program | 20220508072RC | Gang Liu |
| Northeast Normal University | | Gang Liu Xiumei Jiang |
| Fundamental Research Funds for the Central Universities | 135113022 | Gang Liu |
| Fundamental Research Funds for the Central Universities | 135113009 | Xiumei Jiang |

The funders had no role in study design, data collection, and interpretation, or the decision to submit the work for publication.

### Author contributions

Gang Liu, Conceptualization, Formal analysis, Supervision, Funding acquisition, Validation, Investigation, Visualization, Writing – original draft, Project administration, Writing – review and editing; Yunxuan Hou, Conceptualization, Formal analysis, Validation, Investigation, Visualization, Writing – original draft, Writing – review and editing; Xin Jin, Formal analysis, Investigation, Visualization, Methodology; Yixue Zhang, Formal analysis, Validation, Investigation; Chaoyue Sun, Chengquan Huang, Yujie Ren, Jianmin Gao, Formal analysis, Investigation; Xiuli Wang, Software, Investigation, Methodology; Xiumei Jiang, Conceptualization, Formal analysis, Supervision, Funding acquisition, Investigation, Visualization, Writing – original draft, Project administration, Writing – review and editing

### Author ORCIDs

Gang Liu (ID) https://orcid.org/0009-0003-2147-3798

Reviewer #1 (Public Review): https://doi.org/10.7554/eLife.95815.3.sa1
Reviewer #2 (Public Review): https://doi.org/10.7554/eLife.95815.3.sa2
Author response https://doi.org/10.7554/eLife.95815.3.sa3

---

## Additional files

### Supplementary files

• Supplementary file 1. Information for primary antibodies and primers used in this study. Table 1. Primary antibodies used in this study. Table 2. Primers used in the qRT-PCR assays.

• Supplementary file 2. Supplementary Dataset 1 displaying the downregulated DENPs between the EPO+CRISPR NC and EPO+CRISPR HSCB groups of K562 cells.

• Supplementary file 3. Supplementary Dataset 2 displaying the upregulated DENPs between the EPO+CRISPR NC and EPO+CRISPR HSCB groups of K562 cells.

• Supplementary file 4. Supplementary Dataset 3 showing the full lists of FOG1-binding partners in the EPO+CRISPR NC group of K562 cells.

• Supplementary file 5. Supplementary Dataset 4 showing the full lists of FOG1-binding partners in the EPO+CRISPR HSCB group of K562 cells.

• Supplementary file 6. Supplementary Dataset 5 displaying the full list of proteins detected by the tandem mass tag-based mass spectrometry analysis of nuclear fractions of the EPO+CRISPR NC and EPO +CRISPR HSCB groups of K562 cells.

• MDAR checklist

## Data availability

The original nuclear proteomics data are publicly available in the ProteomeXchange platform under the identifier number of PXD041272. The down- and upregulated DENPs are, respectively, summarized in *Supplementary files 2 and 3*. The full lists of FOG1-binding partners in the EPO+CRISPR NC and EPO+CRISPR HSCB groups of K562 cells are, respectively, displayed in *Supplementary files 4 and 5*. The full list of proteins detected by the tandem mass tag-based mass spectrometry analysis of nuclear fractions of the EPO+CRISPR NC and EPO+CRISPR HSCB groups of K562 cells are displayed in *Supplementary file 6*. All Western blotting source data have been provided and more details about experimental protocols can be obtained from the corresponding authors upon reasonable request.

The following dataset was generated:

| Author(s) | Year | Dataset title | Dataset URL | Database and Identifier |
|---|---|---|---|---|
| Liu G, Hou Y, Jin X, Zhang Y, Sun C, Huang C, Ren Y, Gao J, Wang X, Jiang X | 2023 | HSCB KO vs NC K562 cells TMT LC/MS | https://www.ebi.ac.uk/pride/archive/projects/PXD041272 | PRIDE, PXD041272 |

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
