## [Editor Report · eLife assessment]

This **fundamental** work significantly advances our understanding of how FOG1 nuclear localization is regulated during erythropoiesis and megakaryopoiesis, including the role of EPO and MPL/TPO signaling in this process. The authors provide **compelling** evidence using both K562 and CD34+ cells that heat shock cognate B (HSCB) can promote the proteasomal degradation of TACC3 to regulate the nuclear localization of FOG1, and that this function is independent of its role in iron-sulfur cluster (ISC) biogenesis. Together these data will be of interest to the fields of hematopoiesis and cell biology.

---

## [Referee Report · Reviewer #1 (Public Review)]

Summary:

In the paper entitled "PI3K/HSCB axis facilitates FOG1 nuclear translocation to promote erythropoiesis and megakaryopoiesis", the authors sought to determine the role of HSCB, a known regulator of Iron sulfur cluster transfer, in the generation of erythrocytes and megakaryocytes. They utilized a human primary cell model of hematopoietic differentiation to identify a novel mechanism whereby HSCB is necessary for activation of erythroid and megakaryocytic gene expression through regulation of the nuclear localization of FOG-1, a essential transcription co-regulator of the GATA transcription factors. Their work establishes this novel regulatory axis as a mechanism by which cytokine signaling through EPO-R and MPL drives the lineage-specification of hematopoietic progenitors to erythrocytes and megakaryocytes, respectively.

Impact:

The major impact of this work is in a greater understanding of how cytokine signaling through EPO/TPO function to promote lineage specification of hematopoietic stem/progenitor cells. While the major kinase cascades downstream of the EPO/TPO receptors have been elucidated, how those cascades effect gene expression to promote a specific differentiation program is poorly understood. For this work, we now understand that nuclear localization of FOG is a critical regulatory node by which EPO/TPO signaling is required to launch FOG-dependent gene expression. However, these cytokine receptors have many overlapping and redundant targets, so it still remains to be elucidated how signaling through the different receptors promotes divergent gene expression programs. Perhaps similar regulatory mechanisms exist for other lineage-specifying transcription factors.

Strengths:

The authors use two different cellular models of erythroid differentiation (K562 and human primary CD34+ cells) to elucidate the multi-factorial mechanism controlling FOG-1 nuclear localization. The studies are well-controlled and rigorously establish their mechanism through complementary approaches. The differentiation effects are established through cell surface marker expression, protein expression, and gene expression analyses. Novel protein interactions discovered by proteomics analyses were validated through bi-directional co-IP experiments in multiple experimental systems. Protein cellular localization findings are supported by both immunofluorescence and cell fractionation immunoblot analyses. The robustness of their experimental findings gives great confidence for the likelihood that the methods and findings can be reproduced in future work based on their conclusions.

Weaknesses:

The one unexplained step in this intricately described mechanism is how HSCB functions to promote TACC3 degradation. It appears that the proteasome is involved since MG-132 reverses the effect of HSCB deficiency, but no other details are provided. Does HSCB target TACC3 for ubiquitination somehow? Future studies will be required to understand this portion of the mechanism.

One weakness of the study design is that no in vivo experiments are conducted. The authors comment that the HSCB mouse phenotype is too dramatic to permit studies of erythropoiesis in vivo; however, a conditional approach could have been pursued.

It should also be noted that a previous study had already shown that TACC3 regulates the nuclear localization of FOG-1, so this portion of the mechanism is not entirely novel. However, the role of HSCB and the proteasomal degradation of TACC3 is entirely novel to my knowledge.

---

## [Referee Report · Reviewer #2 (Public Review)]

Summary:

In this manuscript, Liu et al. identified an important pathway regulating the nuclear translocation of the key transcriptional factor FOG1 during human hematopoiesis. The authors show that heat shock cognate B (HSCB) can interact with and promote the proteasomal degradation of TACC3, and this function is independent of its role in iron-sulfur cluster biogenesis. TACC3 represses the activity of FOG1 by sequestering it in the cytoplasm. Therefore, HSCB can promote the nuclear translocation of FOG1 through down-regulating TACC3. The authors further show that the phosphorylation of HSCB by PI3K downstream of the EPO signaling pathway is important for its role in regulating the nuclear translocation of FOG1. The data are solid and the manuscript is overall well written. The findings of this manuscript provide important new knowledge to the fields of hematopoiesis and cell biology.

Strengths:

(1) This study uses a multi-pronged approach that combines techniques from a number of fields to convincingly demonstrate the pathway regulating the nuclear translocation of FOG1 during hematopoiesis. The proposed role of each component in the pathway is well supported by solid data.

(2) This work provides important new insights into the function of HSCB, which was known to be an iron-sulfur cluster assembly protein. This study identifies a new role of HSCB and shows that HSCB can regulate the stability of the TACC3 protein, and this cytoplasmic function of HSCB is regulated by protein phosphorylation by PI3K.

(3) The findings of this work open up new directions for research in hematopoiesis and related fields. For example, are there any other TACC3-binding proteins whose subcellular localization are regulated by the presence or absence of TACC3? What is the E3 ligase responsible for the degradation of TACC3? Does this identified mechanism contribute to the sideroblastic anemias observed in HSCB human patients and animal models?

---

## [Author Response]

The following is the authors’ response to the original reviews.

**General responses to the weaknesses of this work:**
The two reviewers mentioned two major weaknesses of this work:(1) The one unexplained step in this intricately described mechanism is how HSCB functions to promote TACC3 degradation. It appears that the proteasome is involved since MG-132 reverses the effect of HSCB deficiency, but no other details are provided. Does HSCB target TACC3 for ubiquitination somehow? Future studies will be required to understand this portion of the mechanism.

We totally agree that the detailed mechanisms through which HSCB promotes TACC3 degradation should be clarified. We tried to find the ubiquitin ligases involved in this regulatory process but could not identify such a key protein so far. We also investigated whether HSCB itself is a ubiquitin ligase but found that the protein does not possess this activity. We therefore consider this weakness another limitation of this research and have added one sentence to the penultimate paragraph of the Discussion section to address this issue.

(2) This study only uses cell models. The significance of this work may be broadened by further studies using animal models.

We totally agree that in vivo models should be adopted to validate the major findings of this study. As we stated in the penultimate paragraph of the Discussion section, we did not have access to biological samples from the patient harboring the HSCB mutation. Additionally, HSCB constitutive knockout mice died during the embryonic stage, while conditional knockout did not cause embryonic death but resulted in almost no erythroid cells in the bone marrow. Therefore, we were not able to further validate our findings in in vivo models.

**Recommendations for the authors:**

**Reviewer #1 (Recommendations For The Authors):**
Figure 3A - Should include FOG1 on the total cell lysate blots to show if total FOG1 is changing or only the cytoplasmic/nuclear ratio. This is shown later but would be good to include here.

We would like to thank the reviewer for the nice suggestion. We have added the blots for total FOG1 to updated Figure 3A as requested.

Figures 3C and 4F - Should include the qPCR results from control cultures on the graphs (EPO + CRISPR NC and shNC, respectively).

We would like to thank the reviewer for the good suggestion. We have added the control groups for all qPCR assays to the updated figures throughout the study.

Figure 4 - The addition of genetic manipulation of TACC3 to confirm its role in the cytoplasmic retention of FOG1 and failed erythroid differentiation in HSCB-deficient cells would strengthen the conclusions of this figure.

We would like to thank the reviewer for the good suggestion. We initially tried to knock down TACC3 expression through siRNAs to confirm its role in the cytoplasmic retention of FOG1. However, we found that siRNAs that worked well in untreated K562 and erythroid progenitor cells as well as several other cell lines had poor efficiency of knocking down gene expression upon HSCB deficiency. This happened not only to siRNAs targeting TACC3, but also to those targeting several other genes. Interestingly, gene overexpression plasmids worked especially well in HSCB-deficient cells. We were not able to explain these phenomena and chose to use an inhibitor of TACC3 to study its functional implications in this research.

Text should be added to discuss the implications of this work for the lineage-specifying function of GATA-1. There are papers by John Crispino and Alan Cantor/Stu Orkin using the FOG-binding mutant of GATA-1 that implicate FOG1-dependent GATA-1 activity as Meg/Ery specifying, whereas FOG1-independent GATA-1 activity promotes mast cell or eosinophil fate. This work suggests that GATA1-expressing myeloid progenitors where FOG1 is kept cytoplasmic (no EPO signaling) would be driven towards the mast cell fate.

We would like to thank the reviewer for the valuable suggestion. We have added a new paragraph in the Discussion section of the updated manuscript to discuss the implication of this work for the lineage-specifying function of GATA-1.

**Reviewer #2 (Recommendations For The Authors):**
Minor comments:(1) In the model provided in Figure 7H, HSCB and FOG1 bind TACC3 simultaneously. However based on the data provided in Figure 6B and other figures, it seems that their interactions are more likely to be mutually exclusive. Is there a possibility that, besides inducing the degradation of TACC3, the binding of HSCB can inhibit the interaction between TACC3 and FOG1?

We would like to thank the reviewer for the insightful comment. According to the data presented in the updated Figure 5F, TACC3 can simultaneously bind with HSCB and FOG1 in E 2-day HSCs. That is why we depict the simultaneous binding pattern in the model provided in Figure 7H. However, we agree that there is a possibility that the binding of HSCB can inhibit the interaction between TACC3 and FOG1 and have mentioned this possibility in the “Phosphorylation of HSCB by PI3K was necessary for its functionalization during human erythropoiesis” subsection of the “Results” section in the updated manuscript.

(2) Whether the decreased TACC3 protein abundance (Figure 5D) during erythroblast differentiation is mainly due to the effect of HSCB. Can silencing of HSCB block this reduction?

We would like to thank the reviewer for the great question. We have analyzed the protein abundance of TACC3 in HSCB-deficient hematopoietic stem cells induced for erythropoiesis for 0, 2 and 4 days and summarized the results as a new Figure 5E. According to the results, TACC3 protein abundance in HSCB-deficient hematopoietic stem cells exhibited no obvious change when the cells were induced for erythropoiesis for 0, 2 and 4 days. These results suggest that the decreased TACC3 protein abundance during early erythroblast differentiation was indeed due to the effect of HSCB. We only investigated the effect of HSCB on TACC3 abundance in early erythroid progenitors because, as shown in Figure 1, HSCB-deficient hematopoietic stem cells stopped differentiation at an early phase of their erythropoiesis. We have also mentioned these data in the “HSCB facilitated FOG1 nuclear translocation by binding with and mediating the proteasomal degradation of TACC3 upon activation of the EPO/EPOR signaling” subsection of the “Results” section in the updated manuscript.

(3) This study shows that HSCB can be phosphorylated by PI3K, and this modification is important for its role in regulating FOG1 distribution. Does the phosphorylation of HSCB also affect its function in ISC biogenesis?

We would like to thank the reviewer for the instructive question. We have analyzed the mitochondrial and cytosolic aconitase activities in wortmannin-treated K562 and E 2-day HSCs and their respective controls. The results have been summarized as a new Figure S5. According to the results, wortmannin treatment did not significantly affect mitochondrial and cytosolic aconitase activities. Therefore, it seems that HSCB phosphorylation does not affect its function in ISC biogenesis. We have also mentioned these data in the “Phosphorylation of HSCB by PI3K was necessary for its functionalization during human erythropoiesis” subsection of the “Results” section in the updated manuscript.

(4) The method of isolation of nuclear fraction needs to be provided in the "Materials and Methods" section.

We would like to thank the reviewer for the thoughtful suggestion. We have added the required information to the “Nuclear proteomics analysis” subsection of the "Materials and Methods" section in the updated manuscript.